# Doubly Robust Alignment for Large Language Models

**Erhan Xu**[*]
Department of Statistics
LSE
London, UK

**Kai Ye**[*]
Department of Statistics
LSE
London, UK

**Hongyi Zhou**[*]
Department of Mathematics
Tsinghua University
Beijing, China

**Luhan Zhu**
School of Design
LCC, UAL
London, UK

**Francesco Quinzan**[†]
Department of Engineering Science
University of Oxford
Oxford, UK

**Chengchun Shi**[†]
Department of Statistics
LSE
London, UK

## Abstract

This paper studies reinforcement learning from human feedback (RLHF) for aligning large language models with human preferences. While RLHF has demonstrated promising results, many algorithms are highly sensitive to misspecifications in the underlying preference model (e.g., the Bradley-Terry model), the reference policy, or the reward function, resulting in undesirable fine-tuning. To address model misspecification, we propose a doubly robust preference optimization algorithm that remains consistent when either the preference model or the reference policy is correctly specified (without requiring both). Our proposal demonstrates superior and more robust performance than state-of-the-art algorithms, both in theory and in practice. The code is available at `https://github.com/DRPO4LLM/DRPO4LLM`

## 1 Introduction

Recent advances in large language models (LLMs) have revolutionized various natural language processing tasks, ranging from text generation to human-AI conversation and more complex reasoning tasks [1–3]. These models are typically trained in two stages. In the pre-training stage, LLMs learn general linguistic patterns and commonsense knowledge from vast, unlabeled text data through autoregressive next-token prediction. However, pretrained models face a critical objective mismatch: while they are optimized for token prediction, real-world deployment requires alignment with complex human values such as helpfulness, honesty and harmlessness [4]. This mismatch calls for an additional post-training stage, aiming at better aligning these pre-trained models with human preference.

The paper studies reinforcement learning from human feedback (RLHF), a post-training paradigm that adapts pre-trained models through reinforcement learning [RL, 5]. The RLHF literature has rapidly expanded in recent years, where existing algorithms can be broadly categorized as reward-based or preference-based (Section 2 for a review). While demonstrating remarkable success in domains including robotics control, video games, and LLMs fine-tuning [see e.g., 6–10], they often suffer from various model misspecifications (see also Table 1 for a summary):

1. **Preference model misspecification**. Most reward-based algorithms rely on the Bradley-Terry [BT, 11] preference model (see Equation 1). However, this model entails various unrealistic assumptions

---

[*]Erhan Xu, Kai Ye, and Hongyi Zhou contributed equally to this paper and are listed in alphabetical order.
[†]Francesco Quinzan and Chengchun Shi are joint senior contributors and are listed in alphabetical order. Address for correspondence: Francesco Quinzan, Ph.D., `francesco.quinzan@eng.ox.ac.uk`; Chengchun Shi, Ph.D., `c.shi7@lse.ac.uk`.

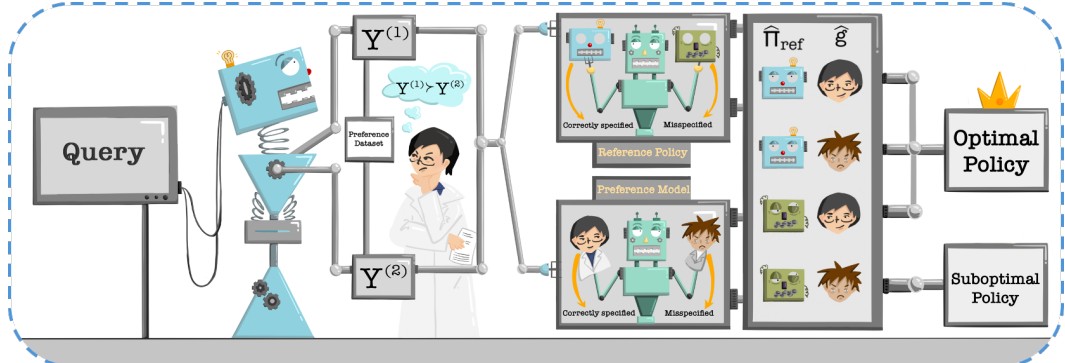

Figure 1: A visualization of our proposed preference optimization algorithm. $\widehat{\pi}_{\mathrm{ref}}$ denotes the specified reference policy whereas $\widehat{g}$ denotes the specified preference model. Our proposal is doubly robust in that it requires correct specification of either the reference policy, or the preference model.

on human preference, including transitivity, context-independence and perfect relationality, which are likely violated based on empirical evidence [12–18]. While some preference-based algorithms impose more general preference model (GPM) assumptions [see e.g., 19], their effectiveness still depends on correct model specification.

2. **Reward model misspecification**. Under the BT model assumption, classical reward-based algorithms first estimate the reward function from human preference data and then apply RL algorithms such as the proximal policy optimization [PPO, 20] to derive the optimal policy. However, policy learning through RL is highly sensitive to the estimated reward. Misspecifying the reward can lead to reward hacking [21, 22] and misguide policy learning [23–25].

3. **Reference policy misspecification**. To alleviate misspecification of the reward, recent algorithms based on direct preference optimization [DPO, 26] propose to express the reward in closed form using the reference policy for policy learning. However, these algorithms are sensitive to the specification of reference policy [27–29].

Drawing from doubly robust estimation methods in econometrics and RL (see Section 2 for a literature review), we introduce a novel RLHF algorithm that is robust to model misspecification and statistically efficient; see Figure 1 for a visualization of our algorithm. Our major contributions are summarized as follows:

- We propose a robust and efficient estimator for preference evaluation, i.e., evaluating the probability of a target policy being preferred over the reference policy. The proposed preference estimator achieves two desirable properties: (i) *double robustness* (Corollary 3) – it converges to the true preference probability when either the preference model or the reference policy is correctly specified, and (ii) *semi-parametric efficiency* (Corollary 4) – it attains the smallest mean squared error (MSE) among all regular and asymptotically linear estimators [30, 31].

- Leveraging this preference estimator, we further develop a preference optimization algorithm for LLM fine-tuning. The proposed algorithm maintains *double robustness* (Corollary (6)) and remains consistent even when the BT model assumption is violated (Theorem 5). Meanwhile, when the BT model assumption holds, its suboptimality gap is likely *smaller* than that of PPO- or DPO-based algorithms (Theorem 7).

## 2 Related Works

Our work is closely related to reward- and preference-based RLHF algorithms, as well as doubly robust (DR) methods. We discuss these related works below.

**Reward-based RLHF.** Reward-based algorithms assume the existence of a latent utility or reward function that determines human preferences, estimate the reward function from the data and apply RL for policy learning. Recent research has focused on addressing practical challenges such as reward

hacking and model-collapse. These issues arise due to PPO's sensitivity to reward specification, gradient clipping thresholds, and the tuning parameter controlling Kullback–Leibler (KL)-divergence regularization [32, 24, 33]. Existing approaches to these challenges fall into three categories: (i) The first category focuses on improving the reward learning algorithm to obtain more accurate reward functions [34–40]. (ii) The second category develops better policy learning algorithms using the estimated reward function [41–47]. (iii) The third category is DPO-based, which bypasses reward learning entirely and directly optimizes policies under the BT model assumption [48–52]. Recent studies have developed robust variants of DPO to handle pairwise noise where preference labels in the training data may be flipped [53–55].

Unlike many of these algorithms, our proposal does not rely on the BT model assumption, and is more robust to the misspecification of reward or reference policy when the BT model holds.

**Preference-based RLHF.** Preference-based algorithms do not assume the existence of a latent reward function at all; instead, they search the optimal policy that maximizes the alignment with human preferences [see e.g., 56]. In particular, there is a growing line of research that adopts the Nash learning from human feedback [NLHF, 57] framework, which formulates the alignment problem as a two-player constant-sum game and solves for policies that achieve the Nash equilibrium [58–64]. Beyond NLHF, [65] develops a Bayesian approach for alignment, whereas [66] and [19] propose energy-based and general preference models to relax the BT model assumption.

Our proposal belongs to this class of preference-based methods. In particular, the proposed algorithm is most closely related to the identity preference optimization (IPO) algorithm proposed by Azar et al. [56], as both maximizing the same objective function in the population level (see Section 4 for the objective). However, unlike IPO, our proposed method is robust to misspecifications of the reference policy. Similarly, compared to [19], the proposed algorithm is more robust to the misspecification of the preference model. Finally, our work differs from NLHF in its primary focus: we study robust and *statistically efficient* preference estimation from data, rather than developing *computationally efficient* algorithms to solve the Nash equilibrium.

**Doubly robust methods.** DR has been extensively studied in statistics, econometrics and machine learning. These methods originate from the missing data and causal inference literature [see e.g., 67, 68]. To illustrate these methods, consider the fundamental causal inference problem of estimating the average treatment effect (ATE) – the difference in the mean outcome between a newly-developed treatment policy and a baseline policy for a given patient population. DR first estimates two models from the data: (i) a propensity score model (similar to the reference policy in LLMs) that characterizes the treatment assignment mechanism and (ii) an outcome regression model (similar to the reward function) that specifies the conditional mean function of a patient's outcome. It then employs both models to construct the ATE estimator, whose consistency requires only one of the models to be correct. Furthermore, when both models are correct, the resulting estimator is semiparametrically efficient [69]. These methods' favorable statistical properties have led to extensive follow-up research [see e.g., 70–92]. A seminal extension appears in Chernozhukov et al. [93], which proposes to learn both the propensity score and outcome regression models using machine learning methods to deal with complex data structures with high-dimensional covariates, texts or images.

Beyond treatment effect estimation in causal inference, doubly robust methods have been widely applied to a broad range of other problems, including the estimation and evaluation of optimal (dynamic) treatment regimes [94–105], conditional independence testing [106–111], offline policy learning [112–117] and off-policy evaluation [OPE, 118–139].

However, none of the aforementioned works considers the application of fine-tuning LLMs – a gap we aim to bridge by connecting these two vibrant research areas. One exception is Lin et al. [140], who considered settings with observed numerical outcome for each answer, rather than the standard RLHF paradigm in which a preference is given comparing two completions.

# 3 RLHF Preliminaries: Data, Modeling and Baseline Algorithms

**Data generating process**. Assume we are given a dataset $\mathcal{D}$, consisting of $n$ i.i.d. tuples of the form $(X, Y^{(1)}, Y^{(2)}, Z)$. Each of these tuples is generated as follows: Given a prompt $X$, two independent responses $(Y^{(1)}, Y^{(2)})$ are generated under a reference policy $\pi_{\text{ref}}$ such that $Y^{(1)}, Y^{(2)} \sim \pi_{\text{ref}}(\bullet|X)$. These data $(X, Y^{(1)}, Y^{(2)})$ are then shown to a human expert, who provides a binary

Table 1: Robustness of different algorithms to model misspecification. Our algorithm is denoted by DRPO, short for doubly robust preference optimization.

| | | Robust to misspecified: | preference model | reward model | reference policy |
|---|---|---|---|---|---|
| RLHF | Reward-based | PPO-based | ✗ | ✗ | ✓ |
| | | DPO-based | ✗ | ✓ | ✗ |
| | Preference-based | IPO [56] | ✓ | - | ✗ |
| | | GPM [19] | ✗ | - | ✓ |
| | | **DRPO** | ✓ | ✓ | ✓ |

preference $Z = \mathbb{I}(Y^{(1)} \succ Y^{(2)})$ where $Y^{(1)} \succ Y^{(2)}$ indicates that the first response is preferred, and $\mathbb{I}(\bullet)$ denotes the indicator function. Additionally, let $g^*$ denote the preference function such that $g^*(X, Y^{(1)}, Y^{(2)}) = \mathbb{P}(Y^{(1)} \succ Y^{(2)}|X)$ determines the probability of $Y^{(1)}$ being favored over $Y^{(2)}$ conditional on $X$.

We remark that the reference policy $\pi_{\text{ref}}$ is not always known. For instance, the responses might be generated by an LLM different from the target model that we wish to fine-tune [8]. Furthermore, the responses might be produced by a heterogeneous set of models rather than a single model [141–143].

**BT model**. As commented in Section 2, most existing reward-based RLHF algorithms impose the BT model assumption, which requires the preference function $g^*$ to take the following form,

$$g^*(x, y^{(1)}, y^{(2)}) = \sigma(r^*(y^{(1)}, x) - r^*(y^{(2)}, x)), \tag{1}$$

where $r^*$ denotes some underlying reward function that measures how well a response answers a given prompt, and $\sigma$ denotes the sigmoid function. As commented in the introduction, this assumption is likely violated due to the inherent intransitivity, inconsistency and stochasticity in human preference.

Assuming (1) holds, the goal is to learn an optimal policy $\pi^*$ that maximizes the expected reward

$$J(\pi) = \mathbb{E}[\mathbb{E}_{y \sim \pi(\bullet|X)} r^*(y, X)], \tag{2}$$

among all policies $\pi$. Here, the outer expectation is taken with respect to the prompt distribution, whereas the inner expectation is taken with respect to the response generated by a given policy $\pi$.

We next introduce two types of baseline algorithms – PPO-based and DPO-based – for learning $\pi^*$. Both approaches operate under Assumption (1).

**PPO-based approaches.** PPO-based algorithms proceed in two steps. In the first step, they compute an estimated reward function $\widehat{r}$ using maximum likelihood estimation or empirical risk minimization. In the second step, they learn $\pi^*$ by maximizing

$$\mathbb{E}_{X \sim \mathcal{D}, \, y \sim \pi(\bullet|X)} \left[\widehat{r}(y, X)\right] - \beta \, D_{\text{KL}} \left[\pi(y \mid X) \, \| \, \pi_{\text{ref}}(y \mid X)\right], \tag{3}$$

over $\pi \in \Pi$ (e.g., a transformer-based policy class), where the expectation is taken over prompts $X$ from the empirical data distribution and responses $y$ from a target policy $\pi$, $D_{\text{KL}}$ denotes the KL divergence measure between the target and reference policies, and the tuning parameter $\beta > 0$ controls the degree to which $\pi$ is allowed to deviate from $\pi_{\text{ref}}$. The KL regularization term in (3) encourages the learned policy to stay close to $\pi_{\text{ref}}$, in order to mitigate over-fitting and prevent the learned policy from collapsing to a narrow set of high-reward responses [24].

**DPO-based approaches**. DPO-based algorithms are motivated by the fact that the argmax to (3) (denoted by $\widehat{\pi}$) can be represented in closed-form using the estimated reward $\widehat{r}$. This in turn yields the following closed-form expression for $\widehat{r}$,

$$\widehat{r}(y, x) = \beta \log \left(\frac{\widehat{\pi}(y|x)}{\pi_{\text{ref}}(y|x)}\right) - C(x), \tag{4}$$

for some response-independent function $C(x)$ that will cancel out in pairwise comparisons. As such, instead of solving $\widehat{\pi}$ in two steps, DPO-based approaches directly parameterize the reward via Equation (4) and compute $\widehat{\pi}$ in a single step – for example, by maximizing the likelihood of the human preference data under the BT model.

To conclude this section, we note that, as shown in Equation (3), the optimal policy computed by PPO can be highly sensitive to the estimated reward function $\widehat{r}$. While DPO-based approaches

eliminate this dependence, Equation (4) reveals that their optimization relies on the specification of the reference policy $\pi_{\text{ref}}$. Due to these sensitivities, even under the idealized setting where the BT model holds, both PPO- and DPO-based algorithms can underperform our proposed algorithm, which is inherently more robust to misspecification in both $\widehat{r}$ and $\pi_{\text{ref}}$. We provide theoretical justification in Section 5 and empirical validation in Section 6.

## 4  Double Robust Preference Evaluation and Optimization

This section introduces the proposed doubly robust approach; see Figure 1 for a visualization. Different from these reward-based algorithms discussed in Section 3, we adopt a preference-based approach that searches the optimal policy by maximizing its total preference. Specifically, given a target policy $\pi$, its *total preference* over the reference policy [56] is defined by

$$p^*(\pi) := \mathbb{P}(\pi \succ \pi_{\text{ref}}) = \mathbb{E}[\mathbb{E}_{y \sim \pi(\bullet|X), y' \sim \pi_{\text{ref}}(\bullet|X)} g^*(X, y, y')],$$

where we recall that $g^*$ denotes the preference function $\mathbb{P}(y > y'|X)$, and the outer expectation is taken with respect to the prompt distribution. As both $Y^{(1)}$ and $Y^{(2)}$ are generated under $\pi_{\text{ref}}$, we have

$$p^*(\pi) = \frac{1}{2} \sum_{a=1}^{2} \mathbb{E}[\mathbb{E}_{y \sim \pi(\bullet|X)} g^*(X, y, Y^{(a)})]. \tag{5}$$

For preference evaluation, our goal is to accurately estimate $p^*(\pi)$ for a given target policy $\pi$ from the dataset $\mathcal{D}$. In the following, we first introduce two baseline estimators: a direct method (DM) estimator and an importance sampling (IS) estimator, where the names are borrowed from the OPE literature [see e.g., 144]. We next introduce our proposed DR estimator, which combines both DM and IS for efficient and robust preference evaluation.

**DM estimator**. The direct method estimator is motivated by (5). It proceeds by first estimating $g^*$ and then plugging the estimated $g^*$ (denoted by $\widehat{g}$) into (5) to construct the estimator,

$$\widehat{p}_{\text{DM}}(\pi) = \frac{1}{2} \mathbb{E}_{X \sim \mathcal{D}, y \sim \pi(\bullet|X)}[\widehat{g}(X, y, Y^{(1)}) + \widehat{g}(X, y, Y^{(2)})], \tag{6}$$

where $X$ is drawn from the empirical data distribution, $y$ is drawn from $\pi$ and the expectation can be approximated using Monte Carlo sampling.

When an external preference model is available, it can be used directly as $\widehat{g}$, as in [57]. Otherwise, $g^*$ can be estimated from the data $\mathcal{D}$. For instance, under the BT model assumption, one can estimate the reward function $r^*$ and plug the estimator into (1) to derive $\widehat{g}$. Alternatively, one can employ more general preference models that do not rely on the BT model.

**IS estimator**. The second baseline estimator is the IS estimator, which is motivated by the following lemma that expresses $p^*(\pi)$ using the IS ratio $w(y, x) = \pi(y|x)/\pi_{\text{ref}}(y|x)$.

**Lemma 1.** *Assume $w(y, x) < \infty$ for any $x$, $y$. Then $p^*(\pi) = \frac{1}{2}\mathbb{E}[w(Y^{(1)}, X)Z + w(Y^{(2)}, X)(1 - Z)]$.*

The proof of Lemma 1 is straightforward. It follows directly from the symmetry of pairwise comparisons where the preference can be equivalently expressed using either $g^*(X, y, y')$ or $1 - g^*(X, y', y)$, and an application of the change-of-measure theorem (see Appendix A.1).

Based on this identity, we define the following IS estimator:

$$\widehat{p}_{\text{IS}}(\pi) = \frac{1}{2} \mathbb{E}_{(X, Y^{(1)}, Y^{(2)}, Z) \sim \mathcal{D}} \left[ \frac{\pi(Y^{(1)}|X)}{\widehat{\pi}_{\text{ref}}(Y^{(1)}|X)} Z + \frac{\pi(Y^{(2)}|X)}{\widehat{\pi}_{\text{ref}}(Y^{(2)}|X)} (1 - Z) \right], \tag{7}$$

where $\widehat{\pi}_{\text{ref}}$ denotes an estimated reference policy. If $\pi_{\text{ref}}$ is known, we can directly use the oracle reference policy. Otherwise, for some external datasets [e.g., 8], well-trained reference models are available and can be used as $\widehat{\pi}_{\text{ref}}$. Finally, when no such external model is available and $\pi_{\text{ref}}$ is unknown, we estimate it from the observed data tuples $(X, Y^{(1)}, Y^{(2)})$ using supervised fine-tuning (SFT).

**DR estimator**. A closer look at Equations (6) and (7) reveals that the DM and IS estimators' consistencies depend crucially on the correct specification of the preference function and the reference

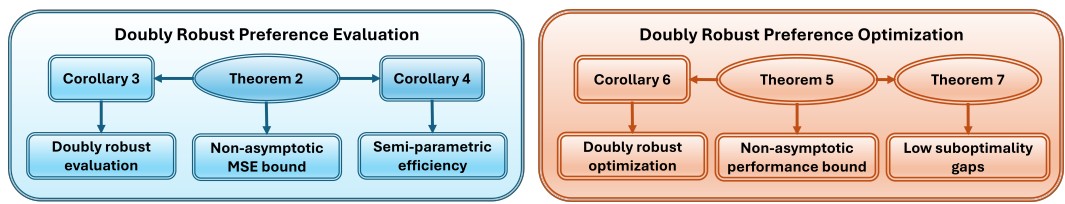

Figure 2: A visualization of our theoretical findings.

policy. We next introduce our proposed DR estimator, which is more robust to misspecifications in these models. It relies on the following estimating function $\psi(X, Y^{(1)}, Y^{(2)}, Z; \pi, \widehat{\pi}_{\text{ref}}, \widehat{g})$, defined as

$$\frac{1}{2} \sum_{a=1}^{2} \mathbb{E}_{y \sim \pi(\bullet|X)}[\widehat{g}(X, y, Y^{(a)})] + \frac{1}{2} \sum_{a=1}^{2} (-1)^{a-1} \frac{\pi(Y^{(a)}|X)}{\widehat{\pi}_{\text{ref}}(Y^{(a)}|X)}[Z - \widehat{g}(X, Y^{(1)}, Y^{(2)})]. \tag{8}$$

By definition, this estimating function contains two terms: (i) the first term is essentially the estimating function of the DM estimator in (6), and (ii) the second term is an augmentation term, which is similar to IS in (7), but with the observed preference $Z$ replaced by its residual $Z - \widehat{g}(X, Y^{(1)}, Y^{(2)})$. The purpose of introducing the additional augmentation term is to correct for the bias introduced by misspecification of the preference model in the DM estimator. This leads to our DR estimator,

$$\widehat{p}_{\text{DR}}(\pi) = \mathbb{E}_{(X, Y^{(1)}, Y^{(2)}, Z) \sim \mathcal{D}} \psi(X, Y^{(1)}, Y^{(2)}, Z; \pi, \widehat{\pi}_{\text{ref}}, \widehat{g}). \tag{9}$$

Similar to the DR estimator in the bandit setting [112], (9) is reduced to the IS estimator when setting $\widehat{g}$ to zero, and the DM estimator when setting the IS ratio $\pi/\widehat{\pi}_{\text{ref}}$ to zero. However, as shown in (8), a key different from those bandit estimators is that in our pairwise comparison setting, each data tuple is used twice – as $(X, Y^{(1)}, Y^{(2)}, Z)$ and $(X, Y^{(2)}, Y^{(1)}, 1 - Z)$ – in constructing the estimating function. This effectively reduces the variance of the resulting estimator. As a result, we will formally show in Section 5 that our DR estimator is semi-parametrically efficient. Additionally, we will establish the consistency of (9) when either $\widehat{g}$ or $\widehat{\pi}_{\text{ref}}$ is correctly specified.

**Preference optimization**. For preference optimization, our goal is to identify the optimal policy that maximizes the average total preference $p^*(\pi)$. Under the BT model assumption, it is immediate to see that the argmax is equivalent to $\pi^*$ defined in (2). Given the proposed DR estimator, we estimate the optimal policy by solving

$$\widehat{\pi} = \arg \max_{\pi \in \Pi} \left\{ \widehat{p}_{\text{DR}}(\pi) - \beta \mathbb{E}_{X \sim \mathcal{D}} D_{\text{KL}}[\pi(\bullet \mid X) \| \widehat{\pi}_{\text{ref}}(\bullet \mid X)] \right\}. \tag{10}$$

We refer to (10) as DRPO, short for doubly robust preference optimization. Theoretically, we will show in Section 5 that our estimated policy $\widehat{\pi}$ achieves a smaller suboptimality gap bound than PPO- and DPO-based algorithms when the BT assumption holds. Practically, we implement three refinements to stabilize the training: (i) clipping the IS ratio to avoid extremely large IS ratio; (ii) designing a pseudo objective function to enable Monte Carlo sampling from the target policy during optimization; (iii) adopting the KL divergence measure from the group relative policy optimization [43] for variance reduction. Details are relegated to Appendix B to save space.

## 5 Theoretical Analysis

We begin with a summary of our theories; Figure 2 outlines the roadmap. Our theories are concerned with (i) the MSE of our preference evaluation estimator $\widehat{p}_{\text{DR}}(\pi)$ (see (9)), and (ii) the performance gap bounds of $\widehat{\pi}$ (see (10)) computed by the proposed preference optimization algorithm. Specifically, Theorem 2 provides a finite sample upper bound for the MSE of $\widehat{p}_{\text{DR}}(\pi)$, which in turn yields its double robustness (Corollary 3) and semi-parametric efficiency (Corollary 4). Meanwhile, Theorem 5 upper bounds the difference in total preference between the optimal in-class policy and $\widehat{\pi}$, without assuming the BT model holds. It reveals the double robustness property of our preference optimization algorithm (Corollary 6). When the BT model holds, Theorem 7 further upper bounds the suboptimal gap of $\widehat{\pi}$, demonstrating that it general achieves smaller gaps than PPO- and DPO-based algorithms.

We next introduce some technical conditions.

**Assumption 1** (Coverage). $\pi/\pi_{\text{ref}}$ and $\pi/\widehat{\pi}_{\text{ref}}$ are upper bounded by some constant $> 0$.

**Assumption 2** (Boundedness). When the BT model holds, both the oracle reward function $r^*$ and its estimator are bounded functions.

**Assumption 3** (Realizability). When the BT model holds, $\pi^*$ that maximizes the expected reward (see (2)) belongs to the parameterized policy class $\Pi$ in (10).

**Assumption 4** (Model complexity). $\Pi$ belongs to the Vapnik–Chervonenkis (VC) type class [145, Definition 2.1] with a finite VC index $v > 0$.

We remark that similar coverage, boundedness and realizability assumptions are commonly imposed in the OPE and RL literature [see e.g., 146, 147, 144]. The VC-class condition is also frequently assumed in statistics and machine learning [see e.g., 148, 149].

**MSE of $\widehat{p}_{\textbf{DR}}(\pi)$.** We next study the statistical properties of the proposed preference estimator $\widehat{p}_{\text{DR}}(\pi)$. Without loss of generality, we also assume both $\widehat{\pi}_{\text{ref}}$ and $\widehat{g}$ (or $\widehat{r}$, in the case where the BT model holds) are obtained from external models independent of $\mathcal{D}$. This condition is mild. Even when such external models are not available and $\widehat{\pi}_{\text{ref}}$ and $\widehat{g}$ are learned internally from $\mathcal{D}$, independence can be preserved using sample-splitting and cross-fitting [93].

**Theorem 2** (MSE). *Under Assumption 1, with $n$ data tuples, the semi-parametric efficiency bound (SEB) for estimating $p^*(\pi)$ is given by $n^{-1}Var(\psi(X, Y^{(1)}, Y^{(2)}, Z; \pi, \pi_{ref}, g^*))$. Additionally, the MSE of our $\widehat{p}_{DR}(\pi)$ equals*

$$SEB + O\left(\frac{1}{n}\|\widehat{g} - g^*\|\right) + O\left(\frac{1}{n}\|\frac{\widehat{\pi}_{\text{ref}}}{\pi_{\text{ref}}} - 1\|\right) + O\left(\|\frac{\widehat{\pi}_{\text{ref}}}{\pi_{\text{ref}}} - 1\|^2 \cdot \|\widehat{g} - g^*\|^2\right), \qquad (11)$$

*where $\|\widehat{\pi}_{\text{ref}}/\pi_{\text{ref}} - 1\|$ and $\|\widehat{g} - g^*\|$ denote the root mean squared errors of $\widehat{\pi}_{\text{ref}}/\pi_{\text{ref}}$ and $\widehat{g}$; see Appendix A.3 for their definitions.*

The first part of Theorem 2 establishes the SEB – the smallest-possible MSE that one can hope for estimating $p^*(\pi)$. The second part upper bounds the excess MSE of our estimator over SEB. Specifically, this excess MSE consists of three parts: the first two are excess variance terms arising from estimation errors in the reference policy and the preference model, while the third is a bias term introduced by these estimation errors. Notably, (i) it can be shown that SEB scales as $O(n^{-1})$; (ii) the two variance terms decrease to zero as the sample size $n$ approaches infinity; (iii) the bias term is a product of the MSEs of $\widehat{\pi}_{\text{ref}}$ and $\widehat{g}$.

Consequently, when either $\widehat{\pi}_{\text{ref}}$ or $\widehat{g}$ is correctly specified, the MSE of $\widehat{p}_{\text{DR}}(\pi)$ converges to zero as $n$ approaches to infinity. This establishes the double robustness property of our estimator, which we state below.

**Corollary 3** (Doubly robust evaluation). *Under Assumption 1, when either $\widehat{\pi}_{\text{ref}}$ or $\widehat{g}$ is correctly specified, the MSE of $\widehat{p}_{DR}(\pi)$ decays to zero as $n$ approaches to infinity.*

We next consider the case where both $\widehat{\pi}_{\text{ref}}$ and $\widehat{g}$ are "approximately" correct in that both root MSEs $\|\widehat{\pi}_{\text{ref}}/\pi_{\text{ref}} - 1\|$ and $\|\widehat{g} - g^*\|$ decay to zero as $n \to \infty$. Since SEB is of the order $O(n^{-1})$, the first two variance terms in (11) decay to zero at a even faster rate than SEB. Meanwhile, when the product $\|\widehat{\pi}_{\text{ref}}/\pi_{\text{ref}} - 1\|\|\widehat{g} - g^*\| = o(n^{-1/2})$, the last bias term in (11) becomes negligible compared to SEB as well. Together, these conditions imply that the MSE of $\widehat{p}_{\text{DR}}(\pi)$ asymptotically matches the SEB, which establishes the semi-parametric efficiency of our estimator. We also remark that conditions similar to $\|\widehat{\pi}_{\text{ref}}/\pi_{\text{ref}} - 1\|\|\widehat{g} - g^*\| = o(n^{-1/2})$ are widely assumed in the literature [see e.g., 93, 103, 82, 150].

**Corollary 4** (Semi-parametric efficiency). *Under Assumption 1, when both $\|\frac{\widehat{\pi}_{\text{ref}}}{\pi_{\text{ref}}} - 1\|$ and $\|\widehat{g} - g^*\|$ decay to zero as $n \to \infty$, and their product is $o(n^{-1/2})$, then $MSE(\widehat{p}_{DR}(\pi))/SEB \to 1$ as $n \to \infty$.*

**Regret of $\widehat{\pi}$.** Next, we derive the statistical properties of the proposed policy $\widehat{\pi}$. When the BT model assumption is violated, we measure the performance gap of a given policy $\pi$ using the gap between the total preference of the best in-class policy and that of $\pi$, i.e., $\text{Gap}(\pi) = \sup_{\pi' \in \Pi} p^*(\pi') - p^*(\pi)$. By definition, a smaller performance gap indicates a better policy.

**Theorem 5** (Performance gap). *Under Assumptions 1 (assuming it holds for any $\pi \in \Pi$) and 4, then*

$$Gap(\widehat{\pi}) = O\left(\beta + \sqrt{\frac{v}{n}} + \frac{v}{n} + \|\widehat{\pi}_{\text{ref}}/\pi_{\text{ref}} - 1\|\|\widehat{g} - g^*\|\right). \qquad (12)$$

It can be seen from (13) that the performance gap depends on several factors: (i) it decays with the sample size $n$; (ii) it increases with the regularization parameter $\beta$ in the KL divergence penalty; (iii) it increases with $v$, which measures the complexity of the policy class; (iv) it decreases with the estimating error of the reference policy and the preference model. Crucially, the last dependence appears as the product $\|\widehat{\pi}_{\text{ref}}/\pi_{\text{ref}} - 1\|\|\widehat{g} - g^*\|$, which enables us to establish the double robustness property in the context of preference optimization.

**Corollary 6** (Doubly robust optimization). *Suppose $\beta \to 0$ as $n \to 0$. Under the conditions in Theorem 5, when either $\widehat{\pi}_{\text{ref}}$ or $\widehat{g}$ is correctly specified, $\text{Gap}(\widehat{\pi})$ decays to zero as $n \to \infty$.*

Finally, we restrict our attention to the ideal setting where the BT model holds and upper bound the suboptimality gap, defined as the difference in the expected reward between the optimal policy $\pi^*$ and our $\widehat{\pi}$, i.e., $J(\pi^*) - J(\widehat{\pi})$.

**Theorem 7** (Suboptimality gap). *Suppose the BT model assumption in* (1) *holds. Under Assumptions 2, 3, and the conditions in Theorem 5, the suboptimality gap of $\widehat{\pi}$ is upper bounded by*

$$O\Big(\beta + \sqrt{\frac{v}{n}} + \frac{v}{n} + \|\widehat{\pi}_{\text{ref}}/\pi_{\text{ref}} - 1\|\|\widehat{r} - r^*\|\Big). \tag{13}$$

*Meanwhile, for PPO-based algorithms, their suboptimality gaps are bounded by*

$$O\Big(\beta + \sqrt{\frac{v}{n}} + \frac{v}{n} + \|\widehat{r} - r^*\|\Big). \tag{14}$$

*Finally, for DPO-based algorithms, their suboptimality gaps are bounded by*

$$O\Big(\exp(-\bar{c}\beta^{-1}) + \beta^{-1}\sqrt{\frac{v}{n}} + \|\widehat{\pi}_{\text{ref}}/\pi_{\text{ref}} - 1\|\Big), \tag{15}$$

*for some constant $\bar{c} > 0$, under conditions specified in Appendix A.7.*

According to (13) and (14) that, by using a sufficiently small $\beta$, the suboptimality gaps of PPO-based and our algorithms are of the order $O(n^{-1/2} + \|\widehat{r} - r^*\|)$ and $O(n^{-1/2} + \|\widehat{\pi}_{\text{ref}}/\pi_{\text{ref}} - 1\|\|\widehat{r} - r^*\|)$, respectively. As for DPO-based algorithms, setting $\beta = \bar{c}^{-1}C \log n$ for some constant $C > 0$ makes the first term in (15) of order $O(n^{-C})$, which can be made arbitrarily small with a sufficiently large $C$. The second term remains of order $O(n^{-1/2})$ up to a logarithmic factor, yielding an overall suboptimality gap of $O(n^{-1/2} \log n + \|\widehat{\pi}_{\text{ref}}/\pi_{\text{ref}} - 1\|)$. Consequently, our algorithm's suboptimality gap is more robust to estimation errors in the reference policy and preference model, as these errors influence our bound only through their product. To the contrary, for PPO- and DPO-based algorithms, these errors affect their suboptimality bounds in the first order. In particular, when these errors converge to zero at a rate of $O(n^{-c})$ for some $0 < c < 1/2$, our algorithm achieves strictly smaller suboptimality bounds than both DPO- and PPO-based algorithms.

To conclude this section, we make two remarks. First, a key novelty of our analysis lies in the derivation of DPO's sub-optimality bounds without relying on linearity assumptions. While there is extensive literature on DPO-based algorithms, their sub-optimality gaps are relatively underexplored. Some recent works derive such bounds under strong linear assumptions, which simplify the analysis by allowing the sub-optimality gap to be expressed directly in terms of parameter estimation error [151]. In contrast, our analysis proceeds without such linear assumptions, which makes the derivation much more challenging. Second, Theorem 7 establishes upper bounds on the sub-optimality gaps, and we discuss the tightness of these bounds in Appendix A.7.

## 6 Experiments

In this section, we first use the IMDb dataset [152] to empirically validate the double robustness property of our preference estimator $\widehat{p}_{\text{DR}}$ (Equation 9) established in Corollary 3. We next compare the proposed preference optimization algorithm (Equation 10) against baseline approaches on the *Too Long; Didn't Read* [TL;DR, 153] and *Anthropic Helpful and Harmless* [HH, 8] datasets.

These datasets are particularly suited for studying preference and/or reference model misspecification. Specifically: (i) TL;DR illustrates reference model misspecification – we use the SFT model trained by `CleanRL` [154], which was learned on a filtered subset of the data, leading to a misspecified

reference policy; (ii) HH illustrates preference model misspecification, as prior works suggest this dataset contains unmodeled pairwise noise beyond BT [53, 54]; (iii) IMDb illustrates both types of misspecification, since it is synthetic dataset where we have access to the ground-truth preference and reference models.

**Preference Evaluation.** We consider the *controlled sentiment generation* task which aims to produce positive movie reviews using the IMDb dataset. We first apply SFT to the `EleutherAI` base model [155], which serves as the reference policy for response generation. The generated responses are then annotated using a pre-trained sentiment classifier to produce preference labels. Using these synthetic data, we train an optimal policy via DPO. Our objective in this section is to evaluate the total preference of this DPO-trained policy over the SFT-based reference policy. Its oracle value, computed via Monte Carlo, is 0.681. Additional details on data generation and model training are provided in Appendix C.1.

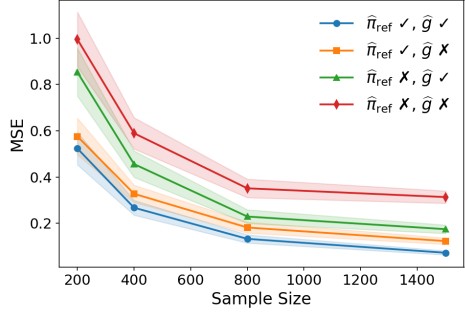

Figure 3: MSEs of different preference evaluation estimators on the IMDb dataset. Shaded areas visualize the 95% confidence bands.

To empirically assess the double robustness property, we evaluate four variants of our preference estimator, each with either the preference model and/or the reference policy correctly specified or misspecified. To misspecify the preference model, we set $\widehat{g}$ to a uniformly random value in $[0, 1]$. To misspecify the reference policy, we use the unfine-tuned `EleutherAI` base model. Figure 3 displays the MSEs (solid lines on left panel) and their associated 95% confidence intervals (shaded areas) of the four estimators across different sample sizes, averaged over 500 simulations. It can be seen that the estimator with both models misspecified (red line) exhibits a significantly larger MSE than the other three and shows minimal improvement beyond 800 samples. To the contrary, when either the preference model or the reference policy is correctly specified (yellow and green lines), the MSE is substantially reduced with a moderately large sample size. This aligns with the double robustness property. Meanwhile, the estimator with both correctly specified models (blue line) achieves the lowest MSE (being very close to zero with 1500 data tuples), supporting its semiparametric efficiency.

**Preference Optimization**. This section considers two tasks: *summarization* and *human dialogue*. First, for summarization, we use the TL;DR dataset, where lengthy Reddit posts serve as prompts and preference annotations are from Stiennon et al. [141], to fine-tune LLMs for concise, informative summaries. Both SFT and reward models for this task are obtained from `CleanRL`. Second, for human dialogue, the HH dataset (human queries as prompts) is used to align LLMs for helpful responses. Since the original SFT and reward models are unavailable, we train them ourselves using the `TRL` framework [156]. For each task, a reward-based BT preference model (using the same reward model for PPO training) and a general preference model [19] are adopted to serve as $\widehat{g}$ (donating as DRPO-BT and DRPO-GPM). Refer to more details of the implementation and baseline model training in Appendix C.2.

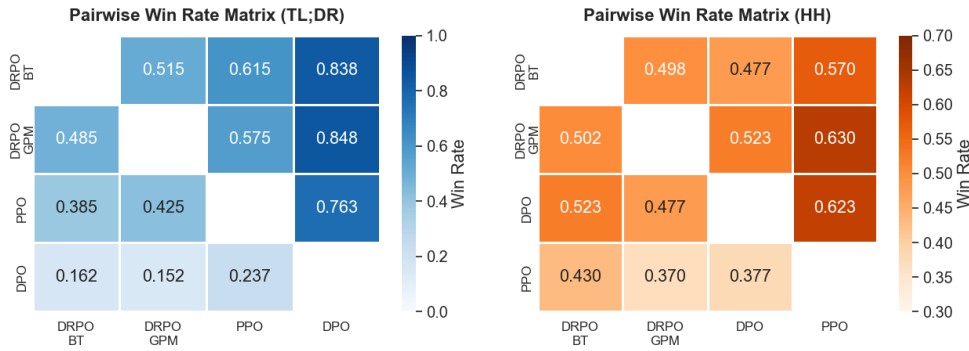

Figure 4: **Pairwise win rate** matrices between different methods across two datasets. **Left:** TL;DR. **Right:** HH. Each entry indicates how often the row method outperforms the column method.

Table 2: Win rates of DRPO (using BT as the preference model) compared to various baseline algorithms on TL;DR. Higher win rates indicate better performance of DRPO over the baseline algorithm.

| Baseline Model | Win Rate (%) |
|---|---|
| DRPO vs Dr. DPO | 72.5 |
| DRPO vs rDPO | 65.0 |
| DRPO vs cDPO | 63.5 |
| DRPO vs CPO | 90.0 |
| DRPO vs ORPO | 57.5 |
| DRPO vs IPO | 98.5 |
| DRPO vs RSO | 69.5 |

Table 3: Win rates of different algorithms compared to SFT on HH. "LC Win Rate" denotes the length-controlled win rate. DRPO uses GPM as preference model.

| Model | LC Win Rate (%) | Win Rate (%) |
|---|---|---|
| Dr. DPO | 92.16 | 90.93 |
| rDPO | 86.89 | 85.71 |
| cDPO | 85.05 | 84.28 |
| CPO | 73.59 | 71.28 |
| ORPO | 75.92 | 53.91 |
| IPO | 78.29 | 78.88 |
| RSO | 80.62 | 79.50 |
| DRPO | 86.38 | 84.84 |

We compare our DRPO against **nine** baseline fine-tuning algorithms, including the standard PPO and DPO, and seven variants of DPO: (i) Dr. DPO [54]; (ii) rDPO [53]; (iii) cDPO [157]; (iv) CPO [158]; (v) ORPO [159]; (vi) IPO [56]; (vii) RSO [48]. Given the absence of ground-truth preference and reward models, we adopt two evaluation strategies. The first strategy uses in-distribution data. Specifically, for both TL;DR and HH, one portion of the dataset is used to fine-tune the LLMs, while the remaining portion is used to generate responses for evaluation. Following prior works [26, 41, 40], we employ GPT-4o-based annotator to compare the quality of responses produced by two LLMs (details in Appendix C.2). Win rates – the percentage of cases in which one LLM's response is preferred over another – are reported at the default temperature of 1.0 in Figure 4 and Table 2, with results at other temperatures provided in Appendix D. The second strategy uses the out-of-distribution data provided via the AlpacaEval 2.0 benchmark [160], which covers a broad collection of human-written instructions designed for general-purpose tasks. Pairwise comparisons are conducted using a GPT-4-Turbo-based annotator. Since summarization is a domain-specific task, we apply the out-of-distribution evaluation only to human dialogue (Table 3).

In *summarization*, both DRPO-BT and DRPO-GPM substantially outperform PPO, DPO (see the left panel of Figure 4), and DPO's variants (Table 2). As mentioned earlier, the reference policy in this dataset is misspecified, likely contributing to the weaker performance of DPO and its variants. Despite trained on the same misspecified reference policy, the superior performance of DRPO highlights its robustness to such misspecification. In *human dialogue*, DRPO-GPM demonstrates the best in-distribution performance, whereas DRPO-BT outperforms PPO and achieves comparable performance to DPO (see the right panel of Figure 4). The poor performance of PPO partly supports the potential misspecification of BT in this task. Despite using the same preference model, DRPO-BT achieves a win rate of 57% against PPO, demonstrating its robustness. As for out-of-distribution evaluation, DRPO performs comparably to robust DPO variants (cDPO, rDPO and Dr. DPO) while attaining higher win rates than other variants (Table 3). As discussed earlier, the HH dataset likely contains pairwise noise, which these robust variants are explicitly designed to handle, whereas DRPO employs a preference model that does not account for such noise. If DRPO were to adopt the same noise-aware preference model used in these methods, its performance would likely improve further.

## 7 Discussion

This work introduces a novel doubly robust preference optimization (DRPO) for LLM fine-tuning. Our approach enables accurate preference evaluation and policy optimization, providing robustness against misspecifications in both the reference policy and the preference model. We formally establish that our preference evaluation estimator is both doubly robust (Corollary 3) and semiparametrically efficient (Corollary 4) and demonstrate that our optimization procedure yields policies with a small performance gap (Theorem 5), and a lower suboptimality bound than DPO and PPO (Theorem 7). Our empirical results reinforce the theoretical advantages, demonstrating DRPO's robustness to reference-policy misspecification (Table 2; Figure 4, left) and preference-model misspecification (Table 3; Figure 4, right).

## Acknowledgments

Hongyi Zhou's research was partially supported by NSFC 12271286 & 11931001 and the China Scholarship Council. Shi and Zhu's research was partially supported by the EPSRC grant EP/W014971/1. Francesco acknowledges funding from ELSA: European Lighthouse on Secure and Safe AI project (grant agreement No. 101070617 under UK guarantee). The authors thank the anonymous referees and the area chair for their insightful and constructive comments, which have led to a significantly improved version of the paper.

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

# Appendix

## A  Technical Proof

In this section, we present the regularity conditions and proofs for all the lemmas and theorems. By nature, the vocabulary size is finite; as such, all random variables – including the prompts $X$ and the responses $Y$ – are discrete. Let the upper bound in Assumption 1 be $\epsilon^{-1}$, where $\epsilon > 0$ is a bounded and fixed constant. Consequently, it is omitted from the error bound in the main text. However, in the proof of Theorems, for completeness, we will explicitly highlight how the leading terms of the error bounds depend on $\epsilon$.

### A.1  Proof of Lemma 1

By direct calculation, it follows that

$$
\begin{aligned}
\mathbb{E}\left\{ w(Y^{(1)}, X) Z \right\} &= \mathbb{E}\left\{ \mathbb{E}\left[ \frac{\pi(Y^{(1)}|X)}{\pi_{\text{ref}}(Y^{(1)}|X)} \mathbb{I}\{Y^{(1)} \succ Y^{(2)}\} \middle| X, Y^{(1)}, Y^{(2)} \right] \right\} \\
&= \mathbb{E}\left\{ \frac{\pi(Y^{(1)}|X)}{\pi_{\text{ref}}(Y^{(1)}|X)} g^*\left(Y^{(1)}, Y^{(2)}, X\right) \right\} \\
&= \mathbb{E}\left\{ \sum_y \pi(y|X) g^*\left(y, Y^{(2)}, X\right) \right\} \\
&= \mathbb{E}\left\{ \mathbb{E}_{y \sim \pi(\bullet|X)} g^*\left(y, Y^{(2)}, X\right) \right\},
\end{aligned}
$$

where the first equality is derived by the law of total expectation, the second equality follows from the definition of the preference function $g^*$, and the third equality follows from the change-of-measure theorem (e.g., Radon–Nikodym theorem).

Following a similar argument and using the fact that $1 - Z = \mathbb{I}(Y^{(2)} \succ Y^{(1)})$, we obtain

$$
\mathbb{E}\left\{ w(Y^{(2)}, X)(1 - Z) \right\} = \mathbb{E}\left\{ \mathbb{E}_{y \sim \pi(\bullet|X)} g^*\left(y, Y^{(1)}, X\right) \right\}.
$$

Consequently, $p^*(\pi) = \frac{1}{2}\mathbb{E}[w(Y^{(1)}, X) Z + w(Y^{(2)}, X)(1 - Z)]$, which finishes the proof of the lemma.

### A.2  Auxiliary lemma for proving Theorem 2

Before proceeding to the proof of Theorem 2, we first introduce an auxiliary lemma.

**Lemma 8.** *Under Assumption 1, with $n$ independent data tuple $W_i = (X_i, Y_i^{(1)}, Y_i^{(2)}, Z_i), i = 1, \ldots n$, the efficient influence function [see e.g., 31, for the detailed definition] for $p^*(\pi)$ is given by $\frac{1}{n}\sum_{i=1}^n \psi(X_i, Y_i^{(1)}, Y_i^{(2)}, Z_i; \pi, \pi_{\text{ref}}, g^*) - p^*(\pi)$, with $\psi$ defined in equation* (8).

*Proof of Lemma 8.* To simplify notation, we denote $\psi(W) = \psi(X, Y^{(1)}, Y^{(2)}, Z; \pi, \pi_{\text{ref}}, g^*)$. Let $\mathcal{M}$ denote the model that generates these data triplets, which are i.i.d. copies of $W = (Z, Y^{(1)}, Y^{(2)}, X)$. This model involves three types of parameters: (i) those to model the probability mass function $f_X(\bullet)$ of the prompt $X$ (denoted by $\gamma$); (ii) those to model the reference policy which generates response $Y^{(1)}, Y^{(2)}$ independently conditional on the prompt $X$ (denoted by $b$) and (iii) those to model the preference probability $g^*$ which characterize the probability of $Y^{(1)}$ is preferred than $Y^{(2)}$ given $X$ (denoted by $\eta$). Then the likelihood function for a data tuple $W$ is given by

$$
l(W; \gamma, b, \eta) = f_\gamma(X) \pi_b(Y^{(1)}|X) \pi_b(Y^{(2)}|X) g_\eta(Y^{(1)}, Y^{(2)}, X)^Z (1 - g_\eta(Y^{(1)}, Y^{(2)}, X))^{1-Z}. \tag{16}
$$

Additionally, let $(\gamma_0, b_0, \eta_0)$ denote the true parameters in the model so that $f_{\gamma_0} = f_X, \pi_{b_0} = \pi_{\text{ref}}$ and $g_{\eta_0} = g^*$.

The proof follows from standard techniques in semi-parametric statistic; see e.g., Chapters 2 & 3 in Bickel et al. [161] and Theorem 3.5 in Tsiatis [31]. See also the proof of Theorem 1 in [124]. Specifically:

1. For any given policy $\pi$, we first prove that $\mathbb{E}[\{\psi(W) - p^*(\pi)\} \nabla \log l(W; \gamma_0, b_0, \eta_0)]$ is a valid derivative of $p^*(\pi)$ with respect to the parameters $(\gamma_0, b_0, \eta_0)$, where $\nabla$ denotes the gradient operator.

2. We next prove that $\psi(W) - p^*(\pi)$ lies in the tangent space of the data generating process model $\mathcal{M}$ (denoted by $\mathcal{T}_{\mathcal{M}}$), that is, $\psi(W) - p^*(\pi) \in \mathcal{T}_{\mathcal{M}}$.

**Step 1:** $\mathbb{E}[\{\psi(W) - p^*(\pi)\} \nabla \log l(W; \gamma_0, b_0, \eta_0)]$ **is a valid derivative of** $p^*(\pi)$ **with respect to** $(\gamma_0, b_0, \eta_0)$.
Noted that the log-likelihood has zero mean. Therefore, in order to prove step 1, we only need to verify the following three equations hold.

(i) $\mathbb{E}\left\{\psi(W)\frac{\partial}{\partial \gamma} \log l(W; \gamma_0, b_0, \eta_0)\right\} = \frac{\partial}{\partial \gamma} p^*(\pi)|_{\gamma=\gamma_0}$,

(ii) $\mathbb{E}\left\{\psi(W)\frac{\partial}{\partial b} \log l(W; \gamma_0, b_0, \eta_0)\right\} = \frac{\partial}{\partial b} p^*(\pi)|_{b=b_0}$,

(iii) $\mathbb{E}\left\{\psi(W)\frac{\partial}{\partial \eta} \log l(W; \gamma_0, b_0, \eta_0)\right\} = \frac{\partial}{\partial \eta} p^*(\pi)_{\eta=\eta_0}$.

By definition, $p^*(\pi)$ can be represented as

$$
\begin{aligned}
p^*(\pi) &= \mathbb{E}[\mathbb{E}_{y_1 \sim \pi_\theta, y_2 \sim \pi_{\text{ref}}} \mathbb{P}(y_1 \succ y_2 | X)] \\
&= \sum_{x, y_1, y_2} g^*(y_1, y_2, x) \pi(y_1|x) \pi_{\text{ref}}(y_2|x) f_X(x).
\end{aligned}
$$

Let $w = (x, y_1, y_2, z)$ denote the realization of $W = (X, Y^{(1)}, Y^{(2)}, Z)$. It follows from equation (16) that

$$
\begin{aligned}
\log l(w; \gamma, b, \eta) = \; &\log f_\gamma(x) + \log \pi_b(y_1|x) + \log \pi_b(y_2|x) \\
&+ z \log g_\eta(y_1, y_2, x) + (1-z) \log(1 - g_\eta(y_1, y_2, x)). \quad (17)
\end{aligned}
$$

With some calculations, we obtain

$$
\begin{aligned}
\frac{\partial}{\partial \gamma} \log l(w; \gamma_0, b_0, \eta_0) &= \frac{1}{f_X(x)} \frac{\partial}{\partial \gamma} f_\gamma(x)\Big|_{\gamma=\gamma_0}, \\
\frac{\partial}{\partial b} \log l(w; \gamma_0, b_0, \eta_0) &= \frac{1}{\pi_{\text{ref}}(y_1|x)} \frac{\partial}{\partial b} \pi_b(y_1|x)\Big|_{b=b_0} + \frac{1}{\pi_{\text{ref}}(y_2|x)} \frac{\partial}{\partial b} \pi_b(y_2|x)\Big|_{b=b_0}, \\
\frac{\partial}{\partial \eta} \log l(w; \gamma_0, b_0, \eta_0) &= \left(\frac{z}{g^*(y_1, y_2, x)} - \frac{1-z}{1 - g^*(y_1, y_2, x)}\right) \frac{\partial}{\partial \eta} g_\eta(y_1, y_2, x)\Big|_{\eta=\eta_0}.
\end{aligned}
$$

In the following proof, we omit $|_{\gamma=\gamma_0}$, $|_{b=b_0}$ and $|_{\eta=\eta_0}$ to ease notation.

**For equation (i):** Let $\text{Ber}(p)$ denote the Bernoulli distribution with success probability $p$. The left-hand-side (LHS) of equation (i) can be represented by

$$
\begin{aligned}
&\mathbb{E}\left\{\psi(W)\frac{\partial}{\partial \gamma} \log l(W; \gamma_0, b_0, \eta_0)\right\} \\
= \; &\frac{1}{2} \sum_{x, y_1, y_2} \mathbb{E}_{z \sim \text{Ber}(g^*(y_1, y_2, x))}\left\{\left(\frac{\pi(y_1|x)}{\pi_{\text{ref}}(y_1|x)} - \frac{\pi(y_2|x)}{\pi_{\text{ref}}(y_2|x)}\right)(z - g^*(y_1, y_2, x))\right. \\
&\qquad\qquad \left. \times \pi_{\text{ref}}(y_1|x) \pi_{\text{ref}}(y_2|x) \frac{\partial}{\partial \gamma} f_\gamma(x)\right\} \\
&+ \frac{1}{2} \sum_{x, y_1, y_2, y^*} (g^*(y^*, y_1, x) + g^*(y^*, y_2, x)) \pi(y^*|x) \pi_{\text{ref}}(y_1|x) \pi_{\text{ref}}(y_2|x) \frac{\partial}{\partial \gamma} f_\gamma(x)
\end{aligned}
$$

Using the fact that $\mathbb{E}_{z \sim \text{Ber}(g^*(y_1, y_2, x))} \{z - g^*(y_1, y_2, x)\} = 0$, the first term on the right-hand-side (RHS) of the above equation vanishes. Therefore,

$$
\begin{aligned}
\mathbb{E}\left\{\psi(W)\frac{\partial}{\partial\gamma}\log l(W;\gamma_0,b_0,\eta_0)\right\} = &\frac{1}{2}\sum_{x,y_1,y^*} g^*(y^*,y_1,x)\pi(y^*|x)\pi_{\text{ref}}(y_1|x)\frac{\partial}{\partial\gamma}f_{\gamma_0}(x) \\
&+ \frac{1}{2}\sum_{x,y_2,y^*} g^*(y^*,y_2,x)\pi(y^*|x)\pi_{\text{ref}}(y_2|x)\frac{\partial}{\partial\gamma}f_{\gamma_0}(x) \\
= &\sum_{x,y,y^*} g^*(y^*,y,x)\pi(y^*|x)\pi_{\text{ref}}(y|x)\frac{\partial}{\partial\gamma}f_{\gamma_0}(x) \\
= &\frac{\partial}{\partial\gamma}p^*(\pi).
\end{aligned}
$$

**For equation (ii)**: Notice that the LHS of equation (ii) can be represented as

$$
\mathbb{E}\left\{\psi(W)\frac{\partial}{\partial b}\log l(W;\gamma_0,b_0,\eta_0)\right\}
$$

$$
\begin{aligned}
= &\frac{1}{2}\sum_{x,y_1,y_2}\mathbb{E}_{z\sim\text{Ber}(g^*(y_1,y_2,x))}\left\{\left(\frac{\pi(y_1|x)}{\pi_{\text{ref}}(y_1|x)} - \frac{\pi(y_2|x)}{\pi_{\text{ref}}(y_2|x)}\right)\left(\frac{1}{\pi_{\text{ref}}(y_1|x)}\frac{\partial}{\partial b}\pi_b(y_1|x) +\right.\right. \\
&\left.\left.\frac{1}{\pi_{\text{ref}}(y_2|x)}\frac{\partial}{\partial b}\pi_b(y_2|x)\right) \times (z - g^*(y_1,y_2,x))\pi_{\text{ref}}(y_1|x)\pi_{\text{ref}}(y_2|x)f_X(x)\right\} \\
&+ \frac{1}{2}\sum_{x,y_1,y_2,y^*}\left(g^*(y^*,y_1,x) + g^*(y^*,y_2,x)\right)\pi(y^*|x)\frac{\partial}{\partial b}[\pi_{b_0}(y_1|x)\pi_{b_0}(y_2|x)]f_X(x).
\end{aligned}
$$

Follows a similar argument in proving equation (i), the first term on the RHS equals zero. The second term can be further represented by

$$
\begin{aligned}
&\frac{1}{2}\frac{\partial}{\partial b}\sum_{x,y_1,y_2,y^*}\left(g^*(y^*,y_1,x) + g^*(y^*,y_2,x)\right)\pi(y^*|x)\pi_{b_0}(y_1|x)\pi_{b_0}(y_2|x)f_X(x) \\
= &\frac{1}{2}\frac{\partial}{\partial b}\sum_{x,y_1,y^*}g^*(y^*,y_1,x)\pi(y^*|x)\pi_{b_0}(y_1|x)f_X(x) \\
&+ \frac{1}{2}\frac{\partial}{\partial b}\sum_{x,y^*,y_2}g^*(y^*,y_2,x)\pi(y^*|x)\pi_{b_0}(y_2|x)f_X(x) \\
= &\sum_{x,y,y^*}g^*(y^*,y,x)\pi(y^*|x)\frac{\partial}{\partial b}\pi_{b_0}(y|x)f_X(x) \\
= &\frac{\partial}{\partial b}p^*(\pi).
\end{aligned}
$$

This finishes the proof of equation (ii).

**For equation (iii)**: Its LHS can be represented as

$$
\mathbb{E}\left\{\psi(w)\frac{\partial}{\partial\eta}\log l(w;\gamma_0,b_0,\eta_0)\right\}
$$

$$
\begin{aligned}
= &\frac{1}{2}\sum_{x,y_1,y_2}\mathbb{E}_{z\sim\text{Ber}(g^*(y_1,y_2,x))}\left\{\left(\frac{\pi(y_1|x)}{\pi_{\text{ref}}(y_1|x)} - \frac{\pi(y_2|x)}{\pi_{\text{ref}}(y_2|x)}\right)(z - g^*(y_1,y_2,x))\right. \\
&\left.\times\pi_{\text{ref}}(y_1|x)\pi_{\text{ref}}(y_2|x)\left(\frac{z}{g^*(y_1,y_2,x)} - \frac{1-z}{1 - g^*(y_1,y_2,x)}\right)\frac{\partial}{\partial\eta}g_\eta(y_1,y_2,x)f_X(x)\right\} \\
&+ \frac{1}{2}\sum_{x,y_1,y_2,y^*}\mathbb{E}_{z\sim\text{Ber}(g^*(y_1,y_2,x))}\left\{\left(g^*(y^*,y_1,x) + g^*(y^*,y_2,x)\right)\pi(y^*|x)\pi_{\text{ref}}(y_1|x)\right. \\
&\left.\times\pi_{\text{ref}}(y_2|x)f_X(x)\left(\frac{z}{g^*(y_1,y_2,x)} - \frac{1-z}{1 - g^*(y_1,y_2,x)}\right)\frac{\partial}{\partial\eta}g_\eta(y_1,y_2,x).\right.
\end{aligned}
$$

The second term is equal to zero due to the fact that

$$\mathbb{E}_{z \sim \text{Ber}(g^*(y_1, y_2, x))} \left\{ \frac{z}{g^*(y_1, y_2, x)} - \frac{1-z}{1 - g^*(y_1, y_2, x)} \right\} = 0.$$

On the other hand, since

$$\mathbb{E}_{z \sim \text{Ber}(g^*(y_1, y_2, x))} \left\{ (z - g^*(y_1, y_2, x)) \left( \frac{z}{g^*(y_1, y_2, x)} - \frac{1-z}{1 - g^*(y_1, y_2, x)} \right) \right\}$$

$$= g^*(y_1, y_2, x) \times (1 - g^*(y_1, y_2, x)) \frac{1}{g^*(y_1, y_2, x)}$$

$$\quad + (1 - g^*(y_1, y_2, x)) \times (-g^*(y_1, y_2, x)) \frac{-1}{1 - g^*(y_1, y_2, x)}$$

$$= 1,$$

the LHS in equation (iii) can be further represented by

$$\frac{1}{2} \sum_{x, y_1, y_2} \left( \frac{\pi(y_1|x)}{\pi_{\text{ref}}(y_1|x)} - \frac{\pi(y_2|x)}{\pi_{\text{ref}}(y_2|x)} \right) \pi_{\text{ref}}(y_1|x) \pi_{\text{ref}}(y_2|x) \frac{\partial}{\partial \eta} g_\eta(y_1, y_2, x) f_X(x)$$

$$= \frac{1}{2} \sum_{x, y_1, y_2} (\pi(y_1|x)\pi_{\text{ref}}(y_2|x) - \pi(y_2|x)\pi_{\text{ref}}(y_1|x)) \frac{\partial}{\partial \eta} g_\eta(y_1, y_2, x) f_X(x)$$

$$= \sum_{x, y_1, y_2} \pi(y_1|x) \pi_{\text{ref}}(y_2|x) \frac{\partial}{\partial \eta} g_\eta(y_1, y_2, x) f_X(x)$$

$$= \frac{\partial}{\partial \eta} p^*(\pi) \tag{18}$$

where the second-to-last equality follows from the fact $\frac{\partial}{\partial \eta} g_\eta(y_1, y_2, x) = -\frac{\partial}{\partial \eta} g_\eta(y_2, y_1, x)$. This finishes the proof of equation (iii).

Thus, with equation (i) - (iii) verified, Step 1 is proven.

**Step 2:** $\psi(W) - p^*(\pi)$ **lies in the tangent space** $\mathcal{T}_\mathcal{M}$.

By definition, the tangent space $\mathcal{T}_\mathcal{M}$ is the linear closure of the set of score functions of the all one-dimensional submodels regarding $\mathcal{M}$ that pass through true parameter; see Definition 2 in [124]. Based on the likelihood function in equation (17), we can explicitly calculate the tangent space of the data generating process model $\mathcal{M}$. In fact, the tangent space $\mathcal{T}_\mathcal{M}$ is a product space, which can be represented as $\mathcal{T}_f \bigoplus \mathcal{T}_\pi \bigoplus \mathcal{T}_g$, with $\mathcal{T}_f, \mathcal{T}_\pi, \mathcal{T}_g$ being the sets of score functions of all one-dimensional submodels passing through the marginal distribution $f_X(x)$, conditional distribution $\pi_{\text{ref}}$ and preference probability $g^*$. Take the calculation of $\mathcal{T}_f$ as an example. Consider a one-dimensional submodel $\{f_\varepsilon(x)\}$, defined as

$$f_\varepsilon(x) = f_X(x)(1 + \varepsilon q(x)),$$

where $q(x)$ satisfies $\sum_x f(x)q^2(x) < \infty$. Since we require $f_\varepsilon$ to be a valid probability mass function, it must satisfy $\sum_x f_\varepsilon(x) = 1$, which indicates $\mathbb{E}q(X) = 0$. Then the score function with respect to $\varepsilon$ is given by

$$\frac{d}{d\varepsilon} \log f_\varepsilon(x) = q(x).$$

Therefore, the tangent space for the marginal distribution function $f(x)$ can be represented as

$$\mathcal{T}_f = \left\{ q(x) : \mathbb{E}[q(X)] = 0, \sum_x f(x)q^2(x) < \infty \right\}.$$

Meanwhile, consider a one-dimensional submodel

$$\pi_\varepsilon(y|x) = \pi_{\text{ref}}(y|x)(1 + \varepsilon q(y, x)),$$

where $q(y, x)$ satisfies $\sum_x q^2(x, y)\pi_{\text{ref}}(y|x) < \infty$. Since we require $\pi_\varepsilon(y|x)$ be a valid conditional probability mass function, it must satisfy $\sum_y \pi_\varepsilon(y|x) = 1$ for any $x$, which indicates $\mathbb{E}_{y \sim \pi_{\text{ref}}} q(y|x) = 0$ for all $x$. Then the score function with respect to $\varepsilon$ is given by

$$\frac{d}{d\varepsilon}\Big|_{\varepsilon=0} \log \pi_\epsilon(y_1|x)\pi_\epsilon(y_2|x) = q(y_1, x) + q(y_2, x).$$

Therefore, the tangent space for the reference policy $\pi_{\mathrm{ref}}$ can be represented as

$$\mathcal{T}_\pi = \left\{ q(y_1, x) + q(y_2, x) : \mathbb{E}_{y \sim \pi_{\mathrm{ref}}}[q(y, x)|X = x] = 0, \sum_y \pi_{\mathrm{ref}}(y|x)q^2(y, x) < \infty \right\}.$$

Following similar arguments, we can obtain

$$\mathcal{T}_g = \left\{ \frac{z - g^*(y_1, y_2, x)}{g^*(1 - g^*)} q(y_1, y_2, x) : \sum_{x, y_1, y_2} q^2(x, y_1, y_2) f(x) \pi_{\mathrm{ref}}(y_1|x) \pi_{\mathrm{ref}}(y_2|x) < \infty \right\}.$$

To verify $\psi(W) - p^*(\pi)$ lies in the tangent space, consider the following three functions:

$$\psi_1(w) := \left( \frac{\pi(y_1|x)}{\pi_{\mathrm{ref}}(y_1|x)} - \frac{\pi(y_2|x)}{\pi_{\mathrm{ref}}(y_2|x)} \right) (z - g^*(y_1, y_2, x)) \pi_{\mathrm{ref}}(y_1|x) \pi_{\mathrm{ref}}(y_2|x) f_X(x)$$

$$= \frac{z - g^*(y_1, y_2, x)}{g^*(1 - g^*)} g^*(1 - g^*) \left( \pi(y_1|x) \pi_{\mathrm{ref}}(y_2|x) - \pi(y_2|x) \pi_{\mathrm{ref}}(y_1|x) \right) f_X(x),$$

$$\psi_2(y_1, y_2, x) := \mathbb{E}_{\substack{y^* \sim \pi}} \{ g(y^*, y_1, x) + g(y^*, y_2, x) \} - 2 \mathbb{E}_{\substack{y \sim \pi_{\mathrm{ref}}(\bullet|x) \\ y^* \sim \pi(\bullet|x)}} \{ g(y^*, y, x) \},$$

$$\psi_3(x) := 2 \mathbb{E}_{\substack{y \sim \pi_{\mathrm{ref}}(\bullet|x) \\ y^* \sim \pi(\bullet|x)}} \{ g(y^*, y, x) \} - 2 p^*(\pi).$$

It is easy to verify that $\psi_1(W) \in \mathcal{T}_g$, $\psi_2(Y^{(1)}, Y^{(2)}, X) \in \mathcal{T}_\pi$ and $\psi_3(X) \in \mathcal{T}_f$. Therefore,

$$\psi(W) - p^*(\pi) = \frac{1}{2} \left( \psi_1(W) + \psi_2(Y^{(1)}, Y^{(2)}, X) + \psi_3(X) \right) \in \mathcal{T}_{\mathcal{M}}.$$

This finishes the proof of Step 2.

With Step 1 and Step 2 verified, together with the fact that $\mathbb{E}\psi(W) = p^*(\pi)$, we obtain that $\psi(W)$ is an efficient influence function. $\square$

### A.3 Proof of Theorem 2

Let $\mathbb{E}_n$ denote the empirical average over the $n$ tuples $(X, Y^{(1)}, Y^{(2)}, Z)$ in the dataset $\mathcal{D}$. Accordingly, our estimator for $p^*(\pi)$ can be represented by $\mathbb{E}_n[\psi(w; \pi, \widehat{\pi}_{\mathrm{ref}}, \widehat{g})]$.

We further define the following norms:

$$\| \widehat{g} - g^* \| = \left( \mathbb{E}\left[ \widehat{g}(Y^{(1)}, Y^{(2)}, X) - g^*(Y^{(1)}, Y^{(2)}, X) \right]^2 \right)^{1/2}$$

$$\left\| \frac{\widehat{\pi}_{\mathrm{ref}}}{\pi_{\mathrm{ref}}} - 1 \right\| = \left( \mathbb{E}\left[ \frac{\widehat{\pi}_{\mathrm{ref}}(Y^{(1)}|X)}{\pi_{\mathrm{ref}}(Y^{(1)}|X)} - 1 \right]^2 \right)^{1/2}.$$

In the proof of this theorem, we assume these norms are bounded. Such a boundedness assumption is automatically satisfied for $\| \widehat{g} - g^* \|$, since both $g^*$ and $\widehat{g}$ are probabilities. These assumptions are to simplify our finite-sample error bound by omitting some higher-order remainder terms, which can be more heavily dependent on the aforementioned norms.

With some calculations, we can show that

$$\mathbb{E}_n \psi(w; \pi, \widehat{\pi}_{\mathrm{ref}}, \widehat{g}) = \mathbb{E}_n \psi(w; \pi, \pi_{\mathrm{ref}}, g^*) + \mathrm{I} + \mathrm{II} + \mathrm{III},$$

where

$$\mathrm{I} = \frac{1}{2} \mathbb{E}_n \left\{ \sum_{a=1}^2 (-1)^a (Z - g^*(X, Y^{(1)}, Y^{(2)})) \left[ \frac{\pi(Y^{(a)}|X)}{\widehat{\pi}_{\mathrm{ref}}(Y^{(a)}|X)} - \frac{\pi(Y^{(a)}|X)}{\pi_{\mathrm{ref}}(Y^{(a)}|X)} \right] \right\},$$

$$\mathrm{II} = \frac{1}{2} \mathbb{E}_n \left\{ \sum_{a=1}^2 \mathbb{E}_{y \sim \pi(\bullet|x)} \left[ \widehat{g}(X, y, Y^{(a)}) - g^*(X, y, Y^{(a)}) \right] \right\}$$

$$+ \frac{1}{2} \mathbb{E}_n \left\{ \sum_{a=1}^2 (-1)^a \frac{\pi(Y^{(a)}|X)}{\pi_{\mathrm{ref}}(Y^{(a)}|X)} [\widehat{g}(X, Y^{(1)}, Y^{(2)}) - g^*(X, Y^{(1)}, Y^{(2)})] \right\},$$

$$\mathrm{III} = \frac{1}{2} \mathbb{E}_n \left\{ \sum_{a=1}^2 (-1)^a [\widehat{g}(X, Y^{(1)}, Y^{(2)}) - g^*(X, Y^{(1)}, Y^{(2)})] \left[ \frac{\pi(Y^{(a)}|X)}{\widehat{\pi}_{\mathrm{ref}}(Y^{(a)}|X)} - \frac{\pi(Y^{(a)}|X)}{\pi_{\mathrm{ref}}(Y^{(a)}|X)} \right] \right\}.$$

From Lemma 8, we know that $\mathbb{E}_n \psi(w; \pi, \pi_{\text{ref}}, g^*)$ is an unbiased estimator for $p^*(\pi)$ with variance equal to SEB. Since both $\widehat{\pi}_{\text{ref}}$ and $\widehat{g}$ are obtained from external models independent of $\mathcal{D}$, analogous to the proof of Lemma 1, we know that the first term I and the second term II have zero means. The third term III is the bias term. Therefore, we obtain the following bias-variance decomposition for $\text{MSE}(\widehat{p}_{\text{DR}})$:

$$\text{MSE}(\widehat{p}_{\text{DR}}(\pi)) = \text{Var}(\mathbb{E}_n \psi(w; \pi, \pi_{\text{ref}}, g^*) + \text{I} + \text{II} + \text{III}) + (\mathbb{E}[\text{III}])^2 \tag{19}$$

Since $g^*$ is bounded by 1, under the coverage assumption (Assumption 1), we obtain that

$$\text{Var}(\mathbb{E}_n \psi(w; \pi, \pi_{\text{ref}}, g^*)) = \frac{1}{n} \text{Var}(\psi(w; \pi, \pi_{\text{ref}}, g^*)) = O\left(\frac{1}{n} \mathbb{E} \frac{\pi^2(Y|X)}{\pi_{\text{ref}}^2(Y|X)}\right)$$
$$= O\left(\frac{1}{n} \sum_y \mathbb{E} \frac{\pi^2(y|X)}{\pi_{\text{ref}}(y|X)}\right) = O\left(\frac{1}{n\epsilon}\right). \tag{20}$$

Similarly, we have

$$\mathbb{E}\text{I}^2 \leq \frac{1}{n} \mathbb{E}\left\{ \left[ \frac{\pi(Y|X)}{\widehat{\pi}_{\text{ref}}(Y|X)} - \frac{\pi(Y|X)}{\pi_{\text{ref}}(Y|X)} \right]^2 \right\}$$
$$\leq \frac{1}{n} \mathbb{E}\left\{ \frac{\pi^2(Y|X)}{\widehat{\pi}_{\text{ref}}^2(Y|X)} \left[ \frac{\widehat{\pi}_{\text{ref}}(Y|X)}{\pi_{\text{ref}}(Y|X)} - 1 \right]^2 \right\} \tag{21}$$
$$= O\left(\frac{1}{n\epsilon^2} \left\| \frac{\widehat{\pi}_{\text{ref}}}{\pi_{\text{ref}}} - 1 \right\|^2\right),$$

and

$$\mathbb{E}\text{II}^2 = O\left(\frac{1}{n\epsilon^2} \|\widehat{g} - g^*\|^2\right), \qquad \mathbb{E}\text{III}^2 = O\left(\frac{1}{n\epsilon^2} \left\| \frac{\widehat{\pi}_{\text{ref}}}{\pi_{\text{ref}}} - 1 \right\|^2\right). \tag{22}$$

By Cauchy-Schwarz inequality, we have for any random variables $U$ and $V$ that $|\text{Cov}(U, V)| \leq \sqrt{\text{Var}(U)\text{Var}(V)}$. It follows that

$$\text{Cov}\left(\mathbb{E}_n \psi(w; \pi, \pi_{\text{ref}}, g^*), \text{I} + \text{III}\right) = O\left(\frac{1}{n\epsilon^{3/2}} \left\| \frac{\widehat{\pi}_{\text{ref}}}{\pi_{\text{ref}}} - 1 \right\|\right),$$
$$\text{Cov}\left(\mathbb{E}_n \psi(w; \pi, \pi_{\text{ref}}, g^*), \text{II}\right) = O\left(\frac{1}{n\epsilon^{3/2}} \|\widehat{g} - g^*\|\right), \tag{23}$$
$$\text{Cov}\left(\text{I} + \text{III}, \text{II}\right) = O\left(\frac{1}{n\epsilon^2} \|\widehat{g} - g^*\| \cdot \left\| \frac{\widehat{\pi}_{\text{ref}}}{\pi_{\text{ref}}} - 1 \right\|\right).$$

Since $\epsilon$ is a constant, the high-order terms $\text{Var}(\text{I})$, $\text{Var}(\text{I})$ and $\text{Var}(\text{III})$ are dominated by the first two terms in (23). Combining equations (20), (21),(22) with (23) yields

$$\text{Var}(\mathbb{E}_n \psi(w; \pi, \pi_{\text{ref}}, g^*) + \text{I} + \text{II} + \text{III}) = \text{SEB} + O\left(\frac{1}{n\epsilon^{3/2}} \|\widehat{g} - g^*\|\right) + O\left(\frac{1}{n\epsilon^{3/2}} \left\| \frac{\widehat{\pi}_{\text{ref}}}{\pi_{\text{ref}}} - 1 \right\|\right). \tag{24}$$

Finally, using Cauchy-Schwarz inequality again, we obtain that

$$\mathbb{E}|\text{III}| = O\left( \mathbb{E}\left\{ (\widehat{g} - g^*)^2(X, Y^{(1)}, Y^{(2)}) \right\}^{1/2} \mathbb{E}\left\{ \left[ \frac{\pi_{\text{ref}}(Y|X)}{\widehat{\pi}_{\text{ref}}^2(Y|X)} - 1 \right]^2 \frac{\pi^2(Y|X)}{\pi_{\text{ref}}^2(Y|X)} \right\}^{1/2}\right)$$
$$= O\left( \frac{1}{\epsilon} \|\widehat{g} - g^*\| \cdot \|\widehat{\pi}_{\text{ref}}/\pi_{\text{ref}} - 1\|\right).$$

Combining (19) with (25), we obtain that

$$\text{MSE}(\widehat{p}_{\text{DR}}(\pi)) = \mathbb{E}\left\{ \mathbb{E}_n \psi(w; \pi, \widehat{\pi}_{\text{ref}}, \widehat{g}) - p^*(\pi) \right\}^2$$
$$= \text{SEB} + O\left(\frac{1}{n\epsilon^{3/2}} \|\widehat{g} - g^*\|\right) + O\left(\frac{1}{n\epsilon^{3/2}} \|\widehat{\pi}_{\text{ref}}/\pi_{\text{ref}} - 1\|\right)$$
$$+ O\left(\frac{1}{\epsilon^2} \|\widehat{\pi}_{\text{ref}}/\pi_{\text{ref}} - 1\|^2 \cdot \|\widehat{g} - g^*\|^2\right).$$

This finishes the proof of Theorem 2.

## A.4 Proofs of Corollaries 3 and 4

The proofs of Corollaries 3 and 4 follow directly from the assertion of Theorem 2.

## A.5 Proof of Theorem 5

Let $\pi^*$ denote the maximizer of $p^*(\pi)$ in the policy class $\Pi$. Throughout the proof, for any policies $\pi_1$ and $\pi_2$, we use a shorthand and write $\mathbb{E}_{X\sim\mathcal{D}}D_{\mathrm{KL}}[\pi_1(\bullet \mid X) \| \pi_2(\bullet \mid X)]$ as $\mathrm{KL}(\pi_1\|\pi_2)$. Since $\widehat{\pi}$ is a maximizer of $\widehat{p}_{\mathrm{DR}}(\pi) - \beta\mathrm{KL}(\pi\|\widehat{\pi}_{\mathrm{ref}})$, we have

$$\widehat{p}_{\mathrm{DR}}(\widehat{\pi}) - \beta\mathrm{KL}(\widehat{\pi}\|\widehat{\pi}_{\mathrm{ref}}) \geq \widehat{p}_{\mathrm{DR}}(\pi^*) - \beta\mathrm{KL}(\pi^*\|\widehat{\pi}_{\mathrm{ref}}).$$

It directly follows that

$$
\begin{aligned}
& p^*(\pi^*) - p^*(\widehat{\pi}) \\
\leq\ & p^*(\pi^*) - \widehat{p}_{\mathrm{DR}}(\pi^*) + \widehat{p}_{\mathrm{DR}}(\widehat{\pi}) - p^*(\widehat{\pi}) + \beta(\mathrm{KL}(\pi^*\|\widehat{\pi}_{\mathrm{ref}}) - \mathrm{KL}(\widehat{\pi}\|\widehat{\pi}_{\mathrm{ref}})) \\
\leq\ & \mathbb{E}\,|p^*(\pi^*) - \widehat{p}_{\mathrm{DR}}(\pi^*)| + \mathbb{E}\,|\widehat{p}_{\mathrm{DR}}(\widehat{\pi}) - p^*(\widehat{\pi})| + O\left(\beta\log^{-1}\epsilon\right) \\
\leq\ & 2\mathbb{E}\sup_{\pi\in\Pi}|p^*(\pi) - \widehat{p}_{\mathrm{DR}}(\pi)| + O\left(\beta\log^{-1}\epsilon\right),
\end{aligned}
\tag{25}
$$

where the second inequality follows from the coverage assumption, which entails that

$$\mathrm{KL}(\pi\|\widehat{\pi}_{\mathrm{ref}}) = \mathbb{E}_{X\sim\mathcal{D}}\mathbb{E}_{y\sim\pi(\bullet|X)}\log\frac{\pi(y|X)}{\widehat{\pi}_{\mathrm{ref}}(y|X)} = O(\log^{-1}\epsilon).$$

Additionally, following the proof of Theorem 2, the bias of the proposed preference evaluation estimator can be upper bounded by

$$\sup_{\pi\in\Pi}|\mathbb{E}[p^*(\pi) - \widehat{p}_{\mathrm{DR}}(\pi)]| = \mathbb{E}\sup_{\pi\in\Pi}|\mathbb{E}_n\psi(w;\pi,\pi_{\mathrm{ref}},g^*) - p^*(\pi)| + O\left(\frac{1}{\epsilon}\|\widehat{g} - g^*\|\cdot\|\frac{\widehat{\pi}_{\mathrm{ref}}}{\pi_{\mathrm{ref}}} - 1\|\right).
\tag{26}$$

It remains to upper bound the empirical process term $\mathbb{E}\sup_{\pi\in\Pi}|\mathbb{E}_n\psi(w;\pi,\pi_{\mathrm{ref}},g^*) - p^*(\pi)|$. Toward that end, we employ Corollary 5.1 in [145]. To invoke this corollary, notice that

1. According to Assumption 4, $\Pi$ is a policy class with VC dimension $v$. Under Assumption 1, it follows from Lemma A.6 in [145] that the function class $\mathcal{F} = \{\psi(\bullet,\pi,\widehat{\pi}_{\mathrm{ref}},\widehat{g})|\pi\in\Pi\}$ also has a VC dimension of $v$.

2. Using the coverage assumption again, the function class $\mathcal{F}$ is uniformly bounded by $O(1/\epsilon)$.

3. The variance $\sup_{f\in\mathcal{F}}\mathrm{Var}(f(W))$ is uniformly bounded by $O(1/\epsilon^2)$.

Consequently, an application of Corollary 5.1 in [145] yields that

$$
\begin{aligned}
\mathbb{E}\sup_{\pi\in\Pi}|\widehat{p}_{\mathrm{DR}}(\pi) - \mathbb{E}[\widehat{p}_{\mathrm{DR}}(\pi)]| &= O\left(\frac{1}{\sqrt{n}}\sqrt{\frac{v}{\epsilon^2}\log^{-1}\epsilon^2} + \frac{v}{n\varepsilon}\log^{-1}\epsilon^2\right) \\
&= O\left(\frac{1}{\epsilon}\sqrt{\frac{v\log^{-1}\epsilon}{n}} + \frac{v\log^{-1}\epsilon}{n\epsilon}\right).
\end{aligned}
$$

Combining equations (25), (26) and (27), we obtain for any $\pi\in\Pi$ that

$$p^*(\pi^*) - p^*(\widehat{\pi}) = O\left(\beta\log^{-1}\epsilon + \frac{1}{\epsilon}\sqrt{\frac{v\log^{-1}\epsilon}{n}} + \frac{v\log^{-1}\epsilon}{n\epsilon} + \frac{1}{\epsilon}\|\widehat{g} - g^*\|\cdot\|\frac{\widehat{\pi}_{\mathrm{ref}}}{\pi_{\mathrm{ref}}} - 1\|\right).$$

This completes the proof of Theorem 5.

## A.6 Proof of Corollary 6

The proof of Corollary 6 follows directly from the assertion of Theorem 5. Before proving Theorem 7, we discuss the tightness of the suboptimality upper bounds derived therein for PPO, DPO, and DRPO. For each algorithm, its gap contains three components:

- A bias term induced by KL-regularization, which is proportional to $\beta$.

- A statistical complexity term of the form $\sqrt{v/n}$, which depends on the sample size $n$ and the complexity measure $v$ of the policy class.

- A reward/reference policy estimation error term.

Since the first term can be made arbitrarily small by choosing a sufficiently small $\beta$, we discuss the tightness of the second and the third terms below. For the second term, our upper bounds for PPO and DPO match the lower bounds developed in [151], indicating their tightness, Finally, our theoretical investigation reveals that under certain settings – e.g., when the reward is linear and the prompt distribution is multivariate Gaussian – the suboptimality gap of PPO depends linearly on the estimation error of the regression coefficient, which itself is proportional to the reward model estimation error. Meanwhile, for DPO, when there is a constant gap between the specified and oracle reference policy, the algorithm suffers from a constant suboptimality gap that will not converge to zero. This demonstrates the tightness of the third term.

### A.7 Proof of Theorem 7

**Suboptimality gap for DRPO:** If the BT assumption holds, we have $g^*(y_1, y_2, x) = \sigma(r^*(y_1, x) - r^*(y_2, x))$ where $\sigma(x) = 1/(1 + e^{-x})$ is the sigmoid function. Since the sigmoid function is monotonically increasing, under the realizability assumption, $\pi^*$ which maximizes $J(\pi)$ also maximizes $p^*(\pi)$. This follows from the classical results on the maximum rank correlation estimator that has been widely studied in the econometrics literature [see e.g., 162, 163]. Therefore,

$$
\begin{aligned}
p^*(\pi^*) - p^*(\widehat{\pi}) &= \mathbb{E}_{y^* \sim \pi^*, \widetilde{y} \sim \widehat{\pi}, y \sim \pi_{\text{ref}}} \{g^*(y^*, y, x) - g^*(\widetilde{y}, y, x)\} \\
&= \mathbb{E}_{y^* \sim \pi^*, \widetilde{y} \sim \widehat{\pi}, y \sim \pi_{\text{ref}}} \{\sigma'(\xi) \left[(r^*(y^*, x) - r^*(y, x)) - (r^*(\widetilde{y}, x) - r^*(y, x))\right]\}^2 \\
&= \mathbb{E}_{y^* \sim \pi^*, \widetilde{y} \sim \widehat{\pi}} \{\sigma'(\xi)(r^*(y^*, x) - r^*(\widetilde{y}, x))\} \\
&\geq C_0(J(\pi^*) - J(\widehat{\pi})),
\end{aligned}
$$

where $C_0$ is some positive constant and $\xi$ is some real number between $r^*(y^*, x) - r^*(y, x)$ and $r^*(\widetilde{y}, x) - r^*(y, x)$. Here, the second equality follows from mean value theorem. The last equality follows from the identity that $\sigma'(x) = \sigma(x)(1 - \sigma(x))$, which is bounded away from zeroo under Assumption 2 that the reward is bounded by some constant. Thus, we obtain $J(\pi^*) - J(\widehat{\pi}) = O(\text{Gap}(\widehat{\pi}))$ and the suboptimality gap for DRPO follows directly from the assertion in Theorem 5.

**Suboptimality gap for PPO-based algorithm:** We begin with some notations. For a given estimated reward $\widehat{r}$, define

- $l(\pi) = \mathbb{E}[\mathbb{E}_{y \sim \pi}\widehat{r}(y, X)] - \beta\text{KL}(\pi \| \pi_{\text{ref}})$,
- $l_n(\pi) = \mathbb{E}_n\mathbb{E}_{y \sim \pi}\widehat{r}(y, X) - \beta\text{KL}(\pi \| \pi_{\text{ref}})$,
- $\widetilde{\pi} = \arg\max_{\pi \in \Pi} l(\pi)$,
- $\widehat{\pi} = \arg\max_{\pi \in \Pi} l_n(\pi)$.

Using the fact that $l(\widetilde{\pi}) \geq l(\pi^*)$ and $l_n(\widehat{\pi}) \geq l_n(\widetilde{\pi})$, we obtain the following upper bound:

$$
\begin{aligned}
J(\pi^*) - J(\widehat{\pi}) &\leq \mathbb{E}\left\{[J(\pi^*) - l(\pi^*)] + [l(\widetilde{\pi}) - l_n(\widetilde{\pi})] + [l_n(\widehat{\pi}) - l(\widehat{\pi})] + [l(\widehat{\pi}) - J(\widehat{\pi})]\right\} \\
&\leq \mathbb{E}\left\{[J(\pi^*) - l(\pi^*)]\right\} + \mathbb{E}\left\{[l(\widehat{\pi}) - J(\widehat{\pi})]\right\} + 2\mathbb{E}\sup_{\pi \in \Pi}\left\{|l(\pi) - l_n(\pi)|\right\}. \quad (27)
\end{aligned}
$$

For the first term, we have

$$
\begin{aligned}
\mathbb{E}\left\{|J(\pi^*) - l(\pi^*)|\right\} &= \mathbb{E}_{y \sim \pi^*}|\widehat{r}(y, X) - r^*(y, X)| + \beta\text{KL}(\pi^* \| \pi_{\text{ref}}) \\
&= \mathbb{E}_{y \sim \pi_{\text{ref}}}\left[\frac{\pi^*(y|X)}{\pi_{\text{ref}}(y|X)}|\widehat{r}(y, X) - r^*(y, X)|\right] + O(\beta\log^{-1}\epsilon) \\
&= O\left(\frac{1}{\sqrt{\epsilon}}\|\widehat{r} - r^*\|\right) + O(\beta\log^{-1}\epsilon), \quad (28)
\end{aligned}
$$

where the last equation follows from Cauchy-Schwarz inequality.

Using a similar argument, we obtain that $\mathbb{E}\left\{|l(\widehat{\pi}) - J(\widehat{\pi})|\right\} = O\left(\frac{1}{\sqrt{\epsilon}}\|\widehat{r} - r^*\| + \beta\log^{-1}\epsilon\right)$.

Finally, under assumption 2, the function class $\mathcal{F} = \left\{ \sum_y \widehat{r}(y, X) \pi(y|X) \middle| \pi \in \Pi \right\}$ is bounded by a constant. Using similar arguments to the proof of Theorem 5, we can employ Corollary 5.1 in [145] to show that

$$\mathbb{E} \sup_{\pi \in \Pi} \{|l(\pi) - l_n(\pi)|\} = O\left(\frac{v}{n} + \sqrt{\frac{v}{n}}\right) + O(\beta \log^{-1} \epsilon). \tag{29}$$

Combining equations (27), (28) and (29), we obtain that

$$J(\pi^*) - J(\widehat{\pi}) = O\left(\beta \log^{-1} \epsilon + \frac{v}{n} + \sqrt{\frac{v}{n}} + \frac{1}{\sqrt{\epsilon}} \|\widehat{r} - r^*\|\right).$$

**Suboptimality gap for DPO-based algorithm:** We need some additional technical conditions to prove the suboptimality gap for DPO-based algorithms. Recall that when BT-model holds, there exists a one-on-one correspondence between the policy and reward model [26]. We further assume

**Assumption 5** (Realizability). The oracle reward $r^*$ lies in the bounded reward function class $\mathcal{R} = \{\beta \log(\pi(y|x)/\pi_{\text{ref}}(y|x)) + \beta Z(x) : \pi \in \Pi\}$ induced by the policy class $\Pi$.

**Assumption 6** (Coverage). Both $\pi_{\text{ref}}$ and $\widehat{\pi}_{\text{ref}}$ are lower bounded by some constant $\epsilon > 0$.

**Assumption 7** (Suboptimality gap for oracle reward). Let $y_x^* = \arg\max_y r^*(y|x)$ and $\bar{y}_x = \arg\max_{y \neq y^*} r^*(y|x)$. There exists a positive constant $\bar{c}$ such that for any $x$,

$$r^*(y_x^*, x) - r^*(\bar{y}_x, x) \geq \bar{c}.$$

Notice that both the realizability and the coverage in Assumptions 5 and 6 differ from those in the main text. Specifically, Assumption 5 imposes the realizability assumption on the oracle reward rather than the optimal policy whereas Assumption 6 is stronger than that in the main text by requiring the denominators of the IS ratios to be strictly positive.

We also redefine the norm $\|\widehat{\pi}_{\text{ref}}/\pi_{\text{ref}} - 1\|$ as

$$\mathbb{E}\left[ \max\left(\frac{\widehat{\pi}_{\text{ref}}(Y^{(1)}|X)}{\pi_{\text{ref}}(Y^{(1)}|X)}, \frac{\pi_{\text{ref}}(Y^{(1)}|X)}{\widehat{\pi}_{\text{ref}}(Y^{(1)}|X)}\right) - 1\right]^2.$$

Notice that this norm is no smaller than the one used in the proposed algorithm.

We next introduce some notations. For a given estimated reference policy $\widehat{\pi}_{\text{ref}}$, any policy $\pi$ induce a reward function

$$r^\pi(y, x) = \beta \log\left(\frac{\pi(y|x)}{\widehat{\pi}_{\text{ref}}(y|x)}\right) + \beta Z(x) \tag{30}$$

Let $l(\pi)$ be the log-likelihood function induced by reward $r^\pi$ and $l^*(\pi)$ be its variant with $\widehat{\pi}_{\text{ref}}$ in the denominator of (30) replaced by the ground truth $\pi_{\text{ref}}$. Denote $\widetilde{\pi} = \arg\max_\pi \mathbb{E}_n l(\pi)$ and $\widehat{\pi} = \arg\max_\pi \mathbb{E} l(\pi)$. It follows that

$$\begin{aligned}
& \mathbb{E}_n l(\widetilde{\pi}) - \mathbb{E}_n l(\widehat{\pi}) - \mathbb{E} l(\widetilde{\pi}) + \mathbb{E} l(\widehat{\pi}) \\
\leq\ & \mathbb{E} l(\widehat{\pi}) - \mathbb{E} l(\widetilde{\pi}) \\
\leq\ & \mathbb{E} l(\widehat{\pi}) - \mathbb{E} l^*(\widetilde{\pi}) + \mathbb{E} l^*(\widetilde{\pi}) - \mathbb{E} l(\widetilde{\pi}) \\
\leq\ & -C_1 \mathbb{E}\|\widehat{r}(y_1, x) - \widehat{r}(y_2, x) - r^*(y_1, x) + r^*(y_2, x)\|_2^2 + \beta^2 C_2 \mathbb{E}\left(\log \frac{\widehat{\pi}_{\text{ref}}(Y^{(1)}|X)}{\pi_{\text{ref}}(Y^{(1)}|X)}\right)^2 \\
\leq\ & -C_1 \sigma^2 + \beta^2 C_2 \|\widehat{\pi}_{\text{ref}}/\pi_{\text{ref}} - 1\|^2, \tag{31}
\end{aligned}$$

where $\sigma^2 = \mathbb{E}\|\widehat{r}(y_1, x) - \widehat{r}(y_2, x) - r^*(y_1, x) + r^*(y_2, x)\|_2^2$, and both $C_1$ and $C_2$ are positive constants because the Hessian matrix is bounded away from zero and infinity, which follows from the boundedness assumption on the reward. Additionally, the last inequality is due to that $x \leq \exp(x) - 1$ for any $x \geq 0$, which entails

$$\begin{aligned}
\mathbb{E}\left(\log \frac{\widehat{\pi}_{\text{ref}}(Y^{(1)}|X)}{\pi_{\text{ref}}(Y^{(1)}|X)}\right)^2 &\leq \mathbb{E}\left[\log \max\left(\frac{\widehat{\pi}_{\text{ref}}(Y^{(1)}|X)}{\pi_{\text{ref}}(Y^{(1)}|X)}, \frac{\pi_{\text{ref}}(Y^{(1)}|X)}{\widehat{\pi}_{\text{ref}}(Y^{(1)}|X)}\right)\right]^2 \\
&\leq \mathbb{E}\left[\max\left(\frac{\widehat{\pi}_{\text{ref}}(Y^{(1)}|X)}{\pi_{\text{ref}}(Y^{(1)}|X)}, \frac{\pi_{\text{ref}}(Y^{(1)}|X)}{\widehat{\pi}_{\text{ref}}(Y^{(1)}|X)}\right) - 1\right]^2
\end{aligned}$$

Moreover, according to Corollary 5.1 in [145], using similar arguments to the proof of Theorem 5 and PPO-based algorithms, we have

$$
\begin{aligned}
\mathbb{E}_n l(\widetilde{\pi}) - \mathbb{E}_n l(\widehat{\pi}) - \mathbb{E}l(\widetilde{\pi}) + \mathbb{E}l(\widehat{\pi}) &\leq 2\mathbb{E}\sup_{\pi \in \Pi}|l(\pi) - \mathbb{E}l_n(\pi)| \\
&\leq O(\sigma\sqrt{\frac{v}{n}} + \frac{v}{n}).
\end{aligned}
\tag{32}
$$

This together with equation (31) yields that $C_1(\sigma - \bar{c}\sqrt{v/n})^2 \leq \bar{c}v/n + \beta^2 C_2\|\widehat{\pi}_{\mathrm{ref}}/\pi_{\mathrm{ref}} - 1\|^2$ for some constant $\bar{c} > 0$, and hence

$$
\sigma = O\left(\sqrt{\frac{v}{n}} + \beta\|\widehat{\pi}_{\mathrm{ref}}/\pi_{\mathrm{ref}} - 1\|_2\right).
\tag{33}
$$

Recall that $\pi^*$ is the true optimal policy, and $\widehat{\pi}$ in this part of the proof denotes DPO's estimated optimal policy. We further define $\widehat{\pi}^*$ as a softmax optimal policy based on the oracle reward function $r^*$

$$
\widehat{\pi}^*(y|x) = \frac{\pi^*(y|x)\exp(\frac{1}{\beta}r^*(y,x))}{\sum_{y'}\pi^*(y'|x)\exp(\frac{1}{\beta}r^*(y',x))}.
$$

With some calculations, it follows that

$$
\begin{aligned}
&J(\pi^*) - J(\widehat{\pi}) \\
=&\mathbb{E}[\mathbb{E}_{y\sim\pi^*}r^*(y,X) - \mathbb{E}_{y\sim\widehat{\pi}}r^*(y,X)] \\
=&\mathbb{E}(\mathbb{E}_{y\sim\pi^*}r^*(y,X) - \mathbb{E}_{y\sim\widehat{\pi}^*}r^*(y,X)) + \mathbb{E}(\mathbb{E}_{y\sim\widehat{\pi}^*}r^*(y,X) - \mathbb{E}_{y\sim\widehat{\pi}}r^*(y,X)),
\end{aligned}
\tag{34}
$$

where the outer expectations are taken with respect to the prompt distribution.

Recall that $y_x^*$ denotes the optimal response to the prompt $x$. The first term $\mathbb{E}[\mathbb{E}_{y\sim\pi^*}r^*(y,X) - \mathbb{E}_{y\sim\widehat{\pi}^*}r^*(y,X)]$ can be upper bounded by

$$
\begin{aligned}
\mathbb{E}r^*(y_X^*,X) - \mathbb{E}[\mathbb{E}_{y\sim\widehat{\pi}^*}r^*(y,X)] &= \mathbb{E}r^*(y_X^*,X) - \mathbb{E}\left\{\frac{\sum_y r^*(y,X)\widehat{\pi}_{\mathrm{ref}}(y|X)\exp\left(\frac{1}{\beta}r^*(y,X)\right)}{\sum_y \widehat{\pi}_{\mathrm{ref}}(y|X)\exp\left(\frac{1}{\beta}r^*(y,X)\right)}\right\} \\
&\leq \mathbb{E}r^*(y_X^*,X) - \mathbb{E}\left\{\frac{r^*(y_X^*,X)\widehat{\pi}_{\mathrm{ref}}(y_X^*|X)\exp\left(\frac{1}{\beta}r^*(y_X^*,X)\right)}{\sum_y \widehat{\pi}_{\mathrm{ref}}(y|X)\exp\left(\frac{1}{\beta}r^*(y,X)\right)}\right\} \\
&= O\left(\frac{1}{\epsilon}\exp\left(-\frac{\bar{c}}{\beta}\right)\right),
\end{aligned}
$$

where the last equality is due to that under Assumptions 6 and 7, the difference between 1 and the ratio $\frac{\widehat{\pi}_{\mathrm{ref}}(y_X^*|X)\exp\left(\frac{1}{\beta}r^*(y_X^*,X)\right)}{\sum_y \widehat{\pi}_{\mathrm{ref}}(y|X)\exp\left(\frac{1}{\beta}r^*(y,X)\right)}$ is of the order $O\left(\frac{1}{\epsilon}\exp\left(-\frac{\bar{c}}{\beta}\right)\right)$, almost surely.

Using mean value theorem, the second term can be bounded by

$$
\mathbb{E}\sum_y |\widehat{\pi}(y|X) - \widehat{\pi}^*(y|X)| \leq \frac{1}{\beta}\mathbb{E}\max_y|\widehat{r}(y,X) - r^*(y,X)| \leq \frac{1}{\beta\sqrt{\epsilon}}\|\widehat{r} - r^*\|_2,
\tag{35}
$$

where the last inequality follows from the fact that

$$
\begin{aligned}
\|\widehat{r} - r^*\|_2 &= \mathbb{E}\{(\widehat{r} - r^*)^2\}^{1/2} \\
&= \mathbb{E}\left\{\sum_y \pi_{\mathrm{ref}}(y|X)(\widehat{r}(y|X) - r^*(y|X))^2\right\}^{1/2} \\
&\geq \sqrt{\epsilon}\mathbb{E}\left\{\sum_y (\widehat{r}(y|X) - r^*(y|X))^2\right\}^{1/2} \\
&\geq \sqrt{\epsilon}\max_y|\widehat{r}(y,X) - r^*(y|X)|.
\end{aligned}
\tag{36}
$$

To complete the proof, it remains to upper bound $\|\widehat{r} - r^*\|_2$ using $\sigma^2$. Recall that $\sigma^2 = \mathbb{E}\left\|\widehat{r}(Y^{(1)}, X) - \widehat{r}(Y^{(2)}, X) - r^*(Y^{(1)}, X) + r^*(Y^{(2)}, X)\right\|_2^2$. Since $Y^{(2)}$ is independent of $Y^{(1)}$ given $X$ and that $\pi_{\text{ref}}$ is lower bounded by $\epsilon > 0$, it follows that

$$\sigma^2 \geq \epsilon \mathbb{E}\left\|\widehat{r}(Y^{(1)}, X) - \widehat{r}(y_0, X) - r^*(Y^{(1)}, X) + r^*(y_0, X)\right\|_2^2,$$

for a fixed $y_0$. Notice that the RHS corresponds to the mean squared error between $\widehat{r}$ and $r^*$, up to a baseline term that is independent of $Y^{(1)}$. Without loss of generality, we can assume this baseline term $r^*(y_0, X) - \widehat{r}(y_0, X)$ is equal to zero without affecting the validity of the proof. This is because the true reward can be redefined as $r^*(\bullet, X) - r^*(y_0, X)$, since it is equivalent up to a function independent of the response. Similarly, the estimated optimal policy $\widehat{\pi}(\bullet|x)$ computed by DPO can be represented using the difference $\widehat{r}(\bullet, x) - \widehat{r}(y_0, x)$, and we can replace $\widehat{r}$ in (35) using this difference. Consequently, we obtain that $\sigma^2 \geq \epsilon\|\widehat{r} - r^*\|^2$ and hence

$$\|\widehat{r} - r^*\| = O\left(\epsilon^{-1/2}\sqrt{\frac{v}{n}} + \beta\epsilon^{-1/2}\|\widehat{\pi}_{\text{ref}}/\pi_{\text{ref}} - 1\|_2\right).$$

Combining this together with equations (33) and (34), we obtain that the regret is upper bounded by

$$O\left(\frac{\exp(-\bar{c}\beta^{-1})}{\epsilon} + \frac{1}{\beta\epsilon}\sqrt{\frac{v}{n}} + \frac{1}{\epsilon}\|\widehat{\pi}_{\text{ref}}/\pi_{\text{ref}} - 1\|\right).$$

The proof is hence completed.

## B  DRPO Algorithm Details and Practical Implementation

This section details our proposed algorithm. Notably, the reference model $\widehat{\pi}_{\text{ref}}$ and the preference model $\widehat{g}$ are pre-trained independently prior to policy optimization. The proposed objective function is defined as

$$\mathcal{J}(\pi_\theta; \widehat{\pi}_{\text{ref}}, \widehat{g}_\eta, \mathcal{D}) = \widehat{p}_{\text{DR}}(\pi) - \beta\mathbb{E}_{X\sim\mathcal{D}}D_{\text{KL}}[\pi(\bullet \mid X)\,\|\,\widehat{\pi}_{\text{ref}}(\bullet \mid X)]. \tag{37}$$

The gradient of $\mathcal{J}(\pi_\theta)$ is given by:

$$\begin{aligned}
\nabla_\theta\mathcal{J}(\pi_\theta) &= \frac{1}{2}\mathbb{E}_{X,Y^{(1)},Y^{(2)}\sim\mathcal{D}}\left\{\sum_{a=1}^{2}\mathbb{E}_{y\sim\pi_\theta(\bullet|X)}\left[\widehat{g}(X, y, Y^{(a)})\nabla_\theta\log\pi_\theta(y|X)\right]\right.\\
&\quad \left. + \sum_{a=1}^{2}(-1)^{a-1}\frac{\nabla_\theta\pi_\theta(Y^{(a)}|X)}{\widehat{\pi}_{\text{ref}}(Y^{(a)}|X)}\left(Z - \widehat{g}(X, Y^{(1)}, Y^{(2)})\right)\right\}\\
&\quad - \beta\nabla_\theta D_{\text{KL}}[\pi_\theta(\bullet \mid X)\,\|\,\widehat{\pi}_{\text{ref}}(\bullet \mid X)]
\end{aligned} \tag{38}$$

Intuitively, the gradient operates as follows: The first term guides the policy to favor responses preferred by the preference model $\widehat{g}$. When $Y^{(1)} \succ Y^{(2)}$, which means $Z = 1$, the second term enhances the likelihood of $Y^{(1)}$ while diminishing the likelihood of $Y^{(2)}$, and vice versa.

The empirical loss function is constructed such that its negative gradient corresponds to $\nabla_\theta\mathcal{J}(\pi_\theta)$ in Equation 38. The direct-method term is approximated using Monte Carlo sampling by drawing several new responses $\mathcal{D}_X^* := \{Y^* \mid Y^* \sim \pi_\theta(\bullet \mid X)\}$ from the current policy $\pi_\theta$ for a given prompt $X$ at each policy update. A k3-type empirical KL divergence is utilized, following [43].

$$\begin{aligned}
\mathcal{L}_{\text{DRPO}} &= -\frac{1}{2}\mathbb{E}_{X,Y^{(1)},Y^{(2)}\sim\mathcal{D}}\left\{\mathbb{E}_{Y^*\sim\mathcal{D}_X^*}\left[\sum_{a=1}^{2}\widehat{g}(Y^*, Y^{(a)}, X)\log\pi_\theta(Y^*|X)\right]\right.\\
&\quad \left. + \sum_{a=1}^{2}(-1)^{a-1}\frac{\pi_\theta(Y^{(a)}|X)}{\pi_{\text{ref}}(Y^{(a)}|X)}\left(Z - \widehat{g}(X, Y^{(1)}, Y^{(2)})\right)\right\}\\
&\quad + \beta\mathbb{E}_{Y^*\sim\mathcal{D}_X^*, X\sim\mathcal{D}}\left[\frac{\widehat{\pi}_{\text{ref}}(Y^* \mid X)}{\pi_\theta(Y^* \mid X)} - 1 - \log\frac{\widehat{\pi}_{\text{ref}}(Y^* \mid X)}{\pi_\theta(Y^* \mid X)}\right]
\end{aligned} \tag{39}$$

Maximization of $\mathcal{J}(\pi_\theta)$ is achieved by minimizing the loss function. In practice, the original offline dataset is augmented to $\mathcal{D}$ to $\widetilde{\mathcal{D}}$ by including swapped pairs (i.e. for $(X, Y^{(1)}, Y^{(2)}, Z)$, we add $(X, Y^{(2)}, Y^{(1)}, 1-Z)$ to $\widetilde{\mathcal{D}}$, simplifying the empirical loss function (39). Furthermore, the importance sampling ratio is clipped, and its calculation is decoupled from the gradient computation. This is achieved by stopping auto-differentiation for the ratio and multiplying the importance sampling term by $\log \pi_\theta$, which shrinks (rather than eliminates) gradients in small $\widehat{\pi}_{\text{ref}}$ regions while maintaining approximate arithmetic equivalence. Consequently, the loss function is reformulated as:

$$
\begin{aligned}
\mathcal{L}_{\text{DRPO}} &= -\frac{1}{2}\mathbb{E}_{X,Y^{(1)},Y^{(2)}\sim\widetilde{\mathcal{D}}}\Bigg\{ \underbrace{\mathbb{E}_{Y^*\sim\mathcal{D}_X^*}\Big[\widehat{g}(Y^*, Y^{(2)}, X)\log\pi_\theta(Y^*|X)\Big]}_{\text{term I}} \\
&\quad + \text{sg}\Bigg(\underbrace{\text{clip}\Big(\frac{\pi_\theta(Y^{(1)}|X)}{\pi_{\text{ref}}(Y^{(1)}|X)}, 1-\epsilon_1, 1+\epsilon_2\Big)\big(Z - \widehat{g}(Y^{(1)}, Y^{(2)}, X)\big)}_{\text{term II}}\Bigg)\log\pi_\theta(Y^{(1)}\mid X)\Bigg\} \\
&\quad + \beta\mathbb{E}_{Y^*\sim\mathcal{D}_X^*, X\sim\widetilde{\mathcal{D}}}\left[\frac{\widehat{\pi}_{\text{ref}}(Y^*\mid X)}{\pi_\theta(Y^*\mid X)} - 1 - \log\frac{\widehat{\pi}_{\text{ref}}(Y^*\mid X)}{\pi_\theta(Y^*\mid X)}\right]
\end{aligned}
\tag{40}
$$

where $\text{sg}(\bullet)$ denotes stop-gradient operation, $\text{clip}(\bullet, a, b)$ signifies clipping values to the interval $[a, b]$, and $\epsilon_1, \epsilon_2$ are hyperparameters defining the clipping range. See full details in Algorithm 1

## C Experiments Implementation details

For the baseline models training, we follow the framework of TRL: *Transformer Reinforcement Learning* [156] and Transformers: *State-of-the-Art Natural Language Processing* [164]. For the general preference model, we follow the framework of general-preference/general-preference-model proposed by Zhang et al. [19]. All models were trained with default hyperparameter configurations unless otherwise specified.

The Preference Evaluation experiments are conducted on a machine equipped with an NVIDIA RTX 6000 Ada GPU and an AMD Ryzen Threadripper PRO 7945WX 12-core CPU. The Preference Optimization experiments are performed on a system with an H20 NVLink GPU and a 20 vCPU Intel(R) Xeon(R) Platinum 8457C processor. AdamW [165] are used as default optimizer.

### C.1 Preference Evaluation Experiment on IMDb

**Oracle Preference Model**. Since the IMDb dataset does not contain human preference labels, we adopt the known sentiment classifier siebert/sentiment-roberta-large-english [166], as a ground-truth reward-based labeler. This classifier will give a score $s(X, Y) = p(\text{positive} \mid X, Y)$, which we convert into a reward signal using the log-odds transformation:

$$
r^*(X, Y) = \log\left(\frac{s(X, Y)}{1 - s(X, Y)}\right).
$$

Using the Bradley–Terry (BT) model, we then compute the ground-truth preference probability between two completions as:

$$
\mathbb{P}^*(Y^{(1)} \succ Y^{(2)}|X) = \sigma(r^*(X, Y^{(1)}) - r^*(X, Y^{(2)})),
$$

where $\sigma(\bullet)$ is the sigmoid function.

**Data Generation and Policy Training Process.** We begin by fine-tuning supervised fine-tuning (SFT) models initialized from two base architectures of different scales: the EleutherAI/gpt-neo-125m [155] and the Qwen/Qwen2.5-7B [167]. Both models are trained for three epochs on 25,000 samples from the IMDb training set. Prompts are constructed by extracting 5-word prefixes from movie reviews. Using the fine-tuned SFT model as the reference policy, we generate pairs of completions for each prompt. Next, we use the oracle preference model to estimate the preference probabilities between each pair of completions. Based on these probabilities, we sample binary

**Algorithm 1** Double Robust Preference Optimization

**Require:** reference policy $\widehat{\pi}_{\text{ref}}$, preference model $\widehat{g}$, offline dataset $\widetilde{\mathcal{D}} = \{X_i, Y_i^{(1)}, Y_i^{(2)}, Z_i\}$, clipping range $[\epsilon_1, \epsilon_2]$, regularization parameter $\beta$, and other hyperparameters, effective batch size $|\mathcal{B}|$, learning rate $\alpha$ and the optimizer, number of Monte Carlo samples $|\mathcal{D}^*|$.

**Ensure:** trained policy $\pi_\theta$

1: **Initialize** policy $\pi_\theta^{(0)}$, total train steps $T = \frac{|\widetilde{\mathcal{D}}|}{|\mathcal{B}|}$. For brevity let the number of training epochs $N = 1$.

2: **for** $t = 1, \ldots, T$ **do**

3:      **for** $i$ in $\mathcal{B}_t := \{(t-1)|\mathcal{B}|, \ldots, t|\mathcal{B}|\}$ **do**

4:          Sample $\mathcal{D}_{X_i}^* = \{Y_j^* \mid Y_j^* \sim \pi_\theta^{(t-1)}(\bullet \mid X_i)\}_{j \in [|\mathcal{D}^*|]}$.

5:          Esitmate term I:

$$\widehat{\text{I}}_i = \frac{1}{|\mathcal{D}_{X_i}^*|} \sum_{Y^* \in \mathcal{D}_{X_i}^*} \widehat{g}(Y^*, Y_i^{(2)}, X_i) \log \pi_\theta^{(t-1)}(Y^*|X_i)$$

6:          Estimate term II:

$$\widehat{\text{II}}_i = \text{clip}\left( \frac{\pi_\theta^{(t-1)}(Y_i^{(1)}|X_i)}{\pi_{\text{ref}}(Y_i^{(1)}|X_i)}, 1 - \epsilon_1, 1 + \epsilon_2 \right) \left( Z - \widehat{g}(Y_i^{(1)}, Y_i^{(2)}, X_i) \right)$$

7:          Estimate KL divergence:

$$\widehat{D}_{\text{KL}_i} = \frac{1}{|\mathcal{D}_{X_i}^*|} \sum_{Y^* \in \mathcal{D}_{X_i}^*} \left( \frac{\widehat{\pi}_{\text{ref}}(Y^* \mid X)}{\pi_\theta(Y^* \mid X)} - 1 - \log \frac{\widehat{\pi}_{\text{ref}}(Y^* \mid X)}{\pi_\theta(Y^* \mid X)} \right)$$

8:          Compute the empirical loss function on the batch:

$$\mathcal{L} = \frac{1}{|\mathcal{B}_t|} \sum_{i \in \mathcal{B}_t} \left\{ -\frac{1}{2} \left[ \widehat{\text{I}}_i + \text{sg}\left( \widehat{\text{II}}_i \right) \log \pi_\theta^{(t-1)}(Y_i^{(1)} \mid X_i) \right] + \beta \widehat{D}_{\text{KL}_i} \right\}$$

9:      **end for**

10:      update $\theta^{(t)}$ with gradient descent and get $\pi_\theta^{(t)}$:

$$\theta^{(t)} = \theta^{(t-1)} - \alpha \nabla_\theta \mathcal{L}$$

11: **end for**

---

preference labels indicating which response is preferred. This synthetic preference dataset is then used to train a target policy using the Direct Preference Optimization (DPO) algorithm over an additional 3 epochs. To quantify the relative preference for the target policy over the reference policy, we adopt a Monte Carlo estimation approach. Specifically, for each of the 25,000 prefixes in the IMDb test set, both the target and reference policies generate a single completion. The oracle preference model is then used to compute the preference probability between the two completions. Aggregating these results, we estimate the overall probability, which is 0.681, that the target policy's outputs are preferred over those of the reference policy.

**Preference Evaluation Process.** We consider two versions of the reference policy estimator $\widehat{\pi}_{\text{ref}}$: a correctly specified version, where $\widehat{\pi}_{\text{ref}}$ corresponds to the SFT model, and a misspecified version, where $\widehat{\pi}_{\text{ref}}$ corresponds to the untrained base model. Similarly, we consider two versions of the preference estimator $\widehat{g}$: a correctly specified version, which uses the oracle preference model, and a misspecified version, where $\widehat{g}$ is drawn uniformly at random from $[0, 1]$. By taking all pairwise combinations of $\widehat{\pi}_{\text{ref}}$ and $\widehat{g}$, we construct four distinct variants of the preference evaluation framework. For the Direct Method (DM) estimator in Equation 6, we apply a Monte Carlo approach by sampling 8 responses from the target policy for each prompt. For the Importance Sampling (IS) estimator in Equation 7, we use a clipping ratio of 100 when $\widehat{\pi}_{\text{ref}}$ is correctly specified and 40 when it is misspecified. In contrast to the clipping ratio used during preference optimization, a larger ratio is adopted here to better demonstrate the double robustness property of our preference evaluation framework. The results based on the `EleutherAI/gpt-neo-125m` model [155] are presented in

Figure 3 in Section 6, while those based on the `Qwen/Qwen2.5-7B` model [167] are summarized in Table 4.

Table 4: MSE of the proposed preference estimator with a 7B base model. The preference model and reference policy can be misspecified or correctly specified.

| Sample size | 500 | 1000 | 2000 | 3000 |
|---|---|---|---|---|
| Both correct | 0.002212 | 0.001160 | 0.000702 | 0.000390 |
| Wrong preference model | 0.024942 | 0.018757 | 0.016763 | 0.016594 |
| Wrong reference model | 0.066897 | 0.021389 | 0.013358 | 0.008383 |
| Both wrong | 0.265155 | 0.069340 | 0.043276 | 0.045954 |

## C.2 Preference Optimization Experiment on Real Data

**Baseline models training**. For the *summarization* task, we adopt models from a group of Hugging Face, `cleanrl`, known for their validated and quality-assured implementations [154]. Specifically, we use `cleanrl/EleutherAI_pythia-1b-deduped__sft__tldr` as both the reference and initial policy model. This SFT policy is trained via token-level supervised fine-tuning on human-written summaries from a filtered TL;DR Reddit dataset [154]. The associated reward model is `cleanrl/EleutherAI_pythia-1b-deduped__reward__tldr`. For PPO training, we search the hyperparameter over the KL coefficient $\beta \in \{0.05, 0.1, 0.2\}$ and select $\beta = 0.05$ based on empirical performance. Notably,we observe that PPO training can experience policy collapse under low-precision, as the value function fails to fit accurately; thus, PPO models are trained under full precision (FP32). In contrast, all our models are trained using bfloat16 (BF16) for improved computational efficiency. To ensure a fair comparison, we set the maximum response length to 128 for all models, providing a consistent basis for assessing summarization quality. For DPO and its variants, we use default hyperparameter setting in `TRL` with BF16 precision. Notably, Dr.DPO had no official `TRL` implementation, so we adapt the loss function in `DPOTrainer` with Dr.DPO's reweighting strategy and use the suggested hyperparameters in [54].

For *human dialogue*, the SFT model is trained from the base model `Qwen/Qwen2.5-1.5B` [167] to better align with the Helpfulness and Harmlessness (HH) dataset. Unlike the summarization SFT model, this version leverages both the preferred (chosen) and non-preferred (rejected) responses from the HH preference dataset. It is trained for 3 epochs. We also train three versions of the reward model, all from the same base model (`Qwen/Qwen2.5-1.5B`) to avoid additional information, corresponding to epochs 1, 2, and 3, as we observe that PPO training in this setting is highly sensitive to the reward model. When the reward model overfits or becomes overly confident, the KL penalty becomes ineffective, and PPO tends to suffer from policy collapse, hacking the reward model by repeating high-reward tokens. To mitigate this issue, we select the reward model from epoch 1, which achieves an evaluation accuracy of 72.1%. We further conduct a hyperparameter search over KL coefficients $\beta \in \{0.05, 0.1, 0.2\}$ and learning rates in $\{$1e-7, 1e-6, 3e-6$\}$. We select a KL coefficient of 0.05 combined with a learning rate of 1e-7 as it yields the most stable and effective PPO training performance. Similar to those in summarization, DPO and its variants are trained with default setting.

**DRPO Implementation** DRPO implementation inherits `transformers.Trainer` class. For DRPO-BT, we compute the rewards for two candidate responses and output the preference probability under the BT framework as $\widehat{g}$. For DRPO-GPM, we directly compute the preference probability using the corresponding general preference model [19]. Although our proposed algorithm allows the use of a more powerful general preference model for estimating $\widehat{g}$, as in [57], we ensure fairness by training all preference models using the same base model and dataset. This avoids introducing any additional information that could bias the comparison. For both tasks, we set the clipping range to $[0.04, 2.5]$, a fairly casual (and wide) specification only to force the IS ratio to not deviate far from 1 and thus not inject too much variance into our estimation. The regularization parameter $\beta$ is set to 0.04, the same as that in the default `trl` implementation for GRPO [43], which also uses k3-type empirical KL divergence. The number of Monte Carlo samples $|\mathcal{D}^*|$ is set to 3 (TL;DR) or 2 (HH). Although more samples may mitigate bias, the effect of adding samples is marginally decreasing (since the convergence rate is $O((n^*)^{-\frac{1}{2}})$). As such, it is proper to choose a parsimonious volume of samples and thus incurring little extra computational cost compared to PPO. Other not-mentioned

hyperparameters are simply set to default values. For further details, please refer to the examples in the codebase.

**Evaluation** For in-distribution evaluation, we compare DRPO with DPO and PPO using GPT-4o-mini to evaluate the quality of generated response of each task. Specifically, for the language model fine-tuned by either baseline or our method, we can sample a response at a certain temperature after it receives a prompt. With the responses of two methods (say A and B), we feed them with a query asking GPT to judge which is more aligned with certain demands. The query template used for TL;DR is shown in Table 5, which tries to avoid GPT's favor of lengthy responses following [40]. The query template used for HH is shown in Table 6, a standard template that is widely adopted by e.g. [26, 41, 40]. It is noteworthy that we randomly shuffle the order of the responses for each query to eliminate the potential bias from the order of the responses.

Here, temperature is the scaler of logits before softmax, which can be used to adjust the output distribution of a certain policy. In general, a temperature less than 1 tends to make kurtosis of the distribution larger (thus more greedy when generating responses), and a temperature larger than 1 generate even more random responses. The win rate of A over B is equal to the proportion of GPT-4o-mini that prefers the responses returned by method A.

For out-of-distribution evaluation in HH dataset, we evaluate our models using the AlpacaEval 2.0 benchmark [160], an LLM-based automatic evaluator designed to assess models' general performance. The prompt set in AlpacaEval 2.0 is derived from AlpacaFarm [168], which contains a broad collection of human-written instructions covering a wide range of general-purpose tasks beyond the Helpful–Harmless (HH) domain. By default, AlpacaEval 2.0 compares each model-generated response against a reference response produced by GPT-4-Turbo, and a GPT-4-Turbo-based annotator determines which of the two is preferred. However, we observed that all fine-tuning algorithms achieved consistently low win rates when evaluated against GPT-4-Turbo references, likely due to the substantial capability gap between GPT-4-Turbo and the fine-tuned models. To ensure a fairer and more interpretable comparison, we therefore replace the reference responses with those generated by the SFT model, allowing AlpacaEval 2.0 to compute the win rate of each fine-tuning algorithm relative to the SFT baseline.

Table 5: Query template for the summarization task.

```
Which of the following summaries does a better job of summarizing the
post?  Strictly follow two criteria when selecting the best summary:
1.  Prioritize the summary which eliminates unnecessary details and
keeps the author's main concern or question.
2.  Prioritize the shorter summary as long as it remains clear and
preserves the main idea.

Post:  <post>

Response A: <response_a>

Response B: <response_b>

FIRST provide a one-sentence comparison of the two summaries, explain-
ing which
you prefer and why.  SECOND, on a new line, state only "A" or "B" to
indicate your
choice.  Your response should use the format:

Comparison:  <one-sentence comparison and explanation>
Preferred:  <''A'' or ''B''>
```

Table 6: Query template for the human dialogue task.

```
For the following query to a chatbot, which response is more helpful?

Query:  <user_query>

Response A: <response_a>

Response B: <response_b>

FIRST provide a one-sentence comparison of the two responses and
explain which you feel is more helpful.  SECOND, on a new line, state
only
``A'' or ``B'' to indicate which response is more helpful.
Your response should use the format:

Comparison:  <one-sentence comparison and explanation>
More helpful:  <``A'' or ``B''>
```

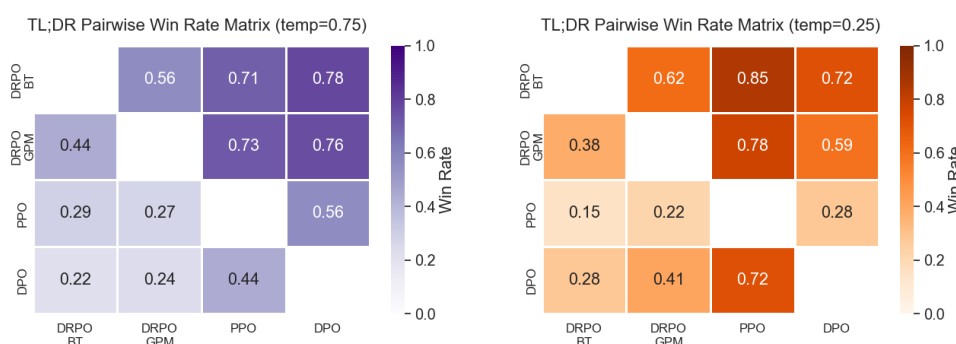

Figure 5: Pairwise Win Rates on TL;DR Dataset under different sampling temperatures (left: 0.75; right: 0.25)

## D    Additional Empirical Results

In this section, we first provide pairwise win rates on the TL;DR dataset with other sampling temperatures (see Figure 5). Our method consistently dominates across all temperatures. DPO's performance improves when temperature gets lower, which is in line with results in [26]. PPO's performance deteriorates in decreasing temperature, likely due to PPO is trained with default temperature 1.0. Next, we present pairwise win rates on HH dataset with other sampling temperatures (see Figure 6). The results are consistent with that of temperature 1.0. In general, DRPO-GPM $\succ$ DRPO-BT $\approx$ DPO $\succ$ PPO, showcasing the robustness of our algorithm.

Additionally, we present some of the sampled responses of our method and baselines and how `gpt-4o-mini` judges the quality of the completions. See Table 7, 8, 9, 10 for TL;DR examples and Table 11, 12, 13, 14 for HH examples.

## E    Limitation and Broader Impact

A potential limitation of our methodology is its reliance on IS ratios for preference evaluation, which can result in high variance when the target and behavior policies differ substantially. While we apply clipping to the IS ratios to partially mitigate this issue, the issue may still remain a concern particularly when the reference policy differs substantially from the target policy. Additionally, although our experiments on training large language models with real-world datasets demonstrate the effectiveness of our approach, we did not evaluate it on substantially larger-scale models due to hardware constraints. This is a potential limitation of our experimental validation.

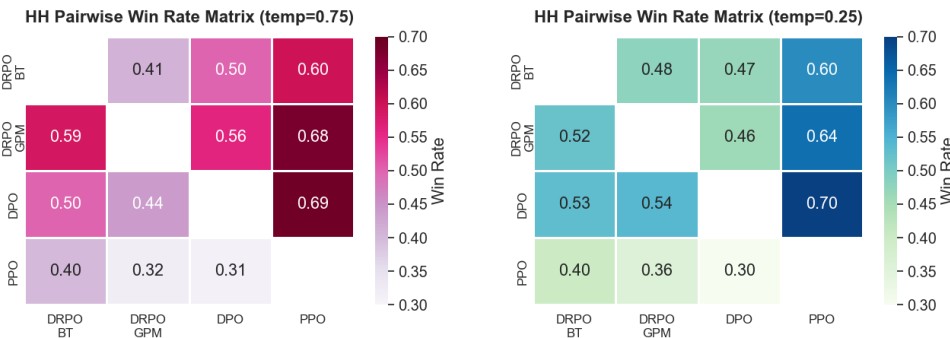

Figure 6: Pairwise Win Rates on HH Dataset under different sampling temperatures (left: 0.75; right: 0.25)

Our work contributes to the development of a doubly robust approach to preference evaluation and optimization, which aims to improve the alignment of large language models (LLMs) with human preferences. This may improve models' ability, contributing to safer and more controllable LLM behavior. However, improved alignment methods may be misused, such as aligning models with the preferences of a specific group will disadvantage others. Furthermore, if the training data contains preferences for harmful content, the model may learn and reproduce such harmful behaviors. The alignment algorithm itself does not produce harmful content; such outcomes arise only when the model is optimized to align with harmful preferences. Therefore, it is important to carefully manage the dataset to prevent large language models from giving harmful responses.

Table 7: Sample Responses of a TL;DR Prompt and Judgement

| Prompt | SUBREDDIT: r/relationships
TITLE: [18/F] My friend [18/F] has naked picture of my boyfriend [17/M]?
POST: This is the weirdest situation ever so please bear with me. My friend was over earlier tonight and she was using my cell phone while I was making food. My cell phone has nudes of my boyfriend on it, I know it's stupid to do that so I don't need a lesson about it please. We've been together for over two years and we fully trust eachother and I'm never doing anything like it again. I had gave her permission to use my phone, she didn't just take it. But I hadn't thought about the pictures, she was just using it to go online on though so I didn't think much of it. I came back in the room and got my phone back and we just went on with our evening. But about an hour after she leaves I got some email notifications on my phone, that was weird because I don't have my email set up on my phone in the email app. So I check it and she had left her email signed in, I saw messages from herself to herself, of pictures that was sent around the time she wouldve been using my phone. I know it's kind of snooping but I clicked on it, and she literally sent pictures of my boyfriend on my phone, to her own email. So she probably has them saved on her computer or something now. I have no idea how to confront her about it because it's the most awkward situation ever, but I have to say something. And I want to make sure she deletes those pictures. I don't even want to continue our friendship anymore if she doesn't. I feel completely betrayed and I think that was disrespectful to my boyfriend, me, our relationship, and our friendship. I haven't told my boyfriend yet as I'm sure he'll be mad at me, but I'll tell him after I resolve this. What can I do?! :(
TL;DR: |
|---|---|
| **DPO** | My friend made sure to send naked pictures to me around the time she was using my phone just to "go online" apparently, there's obviously something suspicious since my boyfriend will be angry at me and how can I prove she doesn't actually have those pictures on her computer now? |
| **DRPO-BT** | my friend has nude pictures of my boyfriend on her cell phone. I was over earlier tonight and she was using my phone to go online. We didn't think much of it because we trust eachother. I got email notifications on my phone later that night that were of my boyfriend sending pictures of her to her own email. I need advice on how to deal with this situation! I don't even want to continue our friendship if it is that bad. |
| **Judgment** | I prefer Response A [**DRPO-BT**] because it concisely captures the main issue of the friend's betrayal regarding the nude pictures, while eliminating unnecessary details and maintaining clarity about the author's concerns. |

Table 8: Sample Responses of a TL;DR Prompt and Judgement

| | |
|---|---|
| **Prompt** | SUBREDDIT: r/tifu
TITLE: TIFU by terrifying a freshie for weeks
POST: So, this is about 4 years ago. I was a sophomore in highschool. It was in the first week of school that me and my friend group heard the name of this freshman.. He had an awesome name. Idk how i can make something comparable.. We'll call him Barry Powers. It wad like that, a superhero type name. So anyway, we heard the name Barry Powers during freshman roll call and we were like "damn! What a sick name!" And made it our ultimate mission to find this guy. And so we searched. It took weeks. Whenever we met other freshmen, we asked, "do you know Barry Powers?" And other questions like "what does he look like?", "where did you last see him?", and "where can we find him?". We really wanted to find this kid and congratulate him on his name. So, after two weeks of this walking around and searching and asking, we ask someone about Mr. Powers and their answer is "oh, he's right there." we get freakin hyped and beeline to the dude, who looks at us with fear in his eyes. Thinking back on it, we had the poor kid, who was eating lunch on his own, in a corner surround by 4 of us. I asked, "hey man. You Barry Powers?" "y-yeah.." He managed to respond. "oh, well, you have a sweet name!" I said. "ok.." Mumbled Powers. Now, the next moment was like the end of an always sunny episode; my friends and I looked around and each other and decided, "oh cool we did it, moving on now." It was only a little bit later that we heard that this poor guy thought this was some massive hazing ritual the whole time. And since we had asked so many people, most of the freshman class thought here was some crazy hazing where we chose one random dude, hunted him down, and did something to him. Oops!
TL;DR: |
| **DPO** | Had the creepy freshies name Barry Powers searched for about four years and finally found an extremely freshie who thought they had some hazing done to it's name. |
| **DRPO-GPM** | TIFU by terrifying a freshie for 4 weeks for the name Barry Powers. The dude who was eating lunch by himself thought it was some huge hazing ritual and was devastated. |
| **Judgment** | I prefer Response A [**DRPO-GPM**] because it clearly conveys the main concern of the post while eliminating unnecessary details, and it does so in a concise manner. |

Table 9: Sample Responses of a TL;DR Prompt and Judgement

| | |
|---|---|
| **Prompt** | SUBREDDIT: r/relationships
TITLE: me (23f) lying to my boyfriend (27) about smoking
POST: So, I've been a smoker since I was 14 years old, I realize at this point how horrible it is for my health and after many failed attempts to quit I seem to be having an extremely hard time. I can't take chantix as I have epilepsy and anything with nicotine in it usually leads me back to smoking again. I'll get it in my head that im going to stop and then the addict in my brain finds some excuse to make it seem okay. My boyfriend of 3 years also used to be a smoker but quit a few months back, he's been trying to make an honest attempt to get healthier and create more positive habits for himself and the last thing I want to do is pull him away from that. He's caught me lying a few times the past couple weeks. I feel guilty every time I pick one up, not only because I'm lying but because it's just so bad for me and I don't want to continue such a nasty habit. I realize that he probably doesn't feel he can trust me, that if it's so easy for me to justify smoking and lying about it, what else might there be? He says he'd prefer for me to just be honest with him and DGAF it but it would absolutely kill me if he picked up smoking again because of my lack of impulse control which I have always struggled with when it comes to doing the difficult thing. I'm afraid of failing and I know that's exactly what im doing by not trying. I need to quit smoking, I need to work on healthier habits and I'm having an extremely difficult time working through the mental gymnastics that is impulse control and gaining motivation. Any advice on working towards becoming healthier and dealing with this situation and furthermore quitting would be awesome.
TL;DR: |
| **PPO** | I've been a smoker since I was 14 years old, I realize at this point how horrible it is for my health and after many failed attempts to quit I seem to be having an extremely hard time. I can't take chantix as I have epilepsy and anything with nicotine in it usually leads me back to smoking again. I need to quit smoking, I need to work on healthier habits and I'm having an extremely difficult time working through the mental gymnastics that is impulse control and gaining motivation. |
| **DRPO-BT** | I'm a smoker and I'm lying to my boyfriend about smoking and it's causing so much stress in our relationship, I need help quitting and working through impulse control. |
| **Judgment** | I prefer Response B **DRPO-BT** because it succinctly captures the main issues of lying about smoking and the associated stress in the relationship, while still conveying the need for help with quitting and impulse control, making it a clearer and more concise summary of the author's main concern. |

Table 10: Sample Responses of a TL;DR Prompt and Judgement

| | |
|---|---|
| **Prompt** | SUBREDDIT: r/relationships
TITLE: Found out my (28f) ex (32m) is having a kid, and now all I want is to sleep around and I'm very depressed.
POST: I dated this guy last year. He impregnated a 20f pretty quickly after we broke up and when I found out, I got extremely sad and jealous, because I wish I was on my way to having a family. We broke up because of infidelity. Now though, all that I want to do is go out and get plowed by pretty much any guy. I don't want a relationship, I just want to go out and have revenge sex. Why is that? I think I'm just so upset, and angry. I don't know what to do, or really how to handle this.
TL;DR: |
| **PPO** | Found out my (28f) ex (32m) is having a kid, and now all I want is to sleep around and I'm very depressed. Why is that? I think I'm just so upset, and angry. I don't know what to do, or really how to handle this. |
| **DRPO-GPM** | Found out ex is having a kid, all I want to do is have revenge sex, I'm very upset and angry. How do I handle this? |
| **Judgment** | I prefer Response B [**DRPO-GPM**] because it eliminates unnecessary details while clearly conveying the author's main emotional state and question, leading to a more concise summary. |

Table 11: Sample Responses of a HH Prompt and Judgement

| | |
|---|---|
| **Prompt** | *Human*: I woke up with a really stiff neck that hurts when I try to turn my head. Can you help me manage the pain?

*Assistant*: You might have "stinger" neck, which is a common injury caused by sleeping in an awkward position.

*Human*: I think that's it. What can I do for the pain?

*Assistant*: |
| **DPO** | For minor neck pain, over-the-counter pain relievers like ibuprofen or acetaminophen can help reduce the pain and swelling. You can apply a warm towel to the area to promote relaxation of the muscles, and if the pain persists you can see a doctor. Are you experiencing any other symptoms? For example, does your neck hurt when you move your head to the side? |
| **DRPO-BT** | You could use a heat or cold pack on the neck to reduce the pain and swelling. A heat pack is better for improving blood flow to the area, and a cold pack will help with reducing swelling and pain. You might feel better in 10-20 minutes. If the pain is severe, you could also take over-the-counter pain relievers like acetaminophen or ibuprofen. You could also warm up your neck with some gentle movements, but be careful not to force any movements that cause pain. |
| **Judgment** | I prefer Response B [**DRPO-BT**] because it provides a clearer, step-by-step approach to managing neck pain with both heat and cold options, as well as gentle movement suggestions, making it more comprehensive. |

Table 12: Sample Responses of a HH Prompt and Judgement

| | |
|---|---|
| **Prompt** | *Human*: What are some ways that I can get smoother skin? |
| | *Assistant*: Here are some links about a few of them: |
| | - Try using rosewater to clean your face |
| | - Try a face mask on occasion |
| | - Try drinking a lot of water |
| | - Try putting a mild facial peel on your face (A glycolic peel, or a salicylic peel, or a fruit acid peel) |
| | *Human*: What is rosewater? |
| | *Assistant*: Rosewater is a special distilled form of rose petals. It's used for beauty products. Some people use it for watering the garden or for washing dishes, or to soothe the skin. |
| | *Human*: How much water should I try drinking for smoother skin? |
| | *Assistant*: |
| **DPO** | It depends on you, but generally you want to drink as much water as you can. |
| | Drinking enough water helps your body flush out toxins. |
| | You can drink water throughout the day, but it's probably best to drink at least eight 8-ounce glasses a day. |
| **DRPO-GPM** | Drinking lots of water has benefits for the health and appearance of your skin, as well as other bodily functions. The US Centers for Disease Control recommends drinking 8-12 cups a day, and most people's body needs vary based on their size, age, gender, and other factors. |
| **Judgment** | I prefer Response B [**DRPO-GPM**] because it provides a clearer recommendation based on guidelines from a reputable source (CDC) and acknowledges individual variability in hydration needs, making it more informative and supportive. |

Table 13: Sample Responses of a HH Prompt and Judgement

| | |
|---|---|
| **Prompt** | *Human*: How do I groom a cat? |
| | *Assistant*: Grooming a cat involves combing their fur with a brush. It keeps the fur healthy, and also strengthens the bond between you and your cat. |
| | *Human*: That sounds good. Are there special brushes for cats? |
| | *Assistant*: |
| **PPO** | Yes, there are many types of brushes for cats, including ones that have soft bristles for easy grooming and ones that have a firmer, more efficient design for cleaning and brushing. Many cat owners like to use a finger brush, which can be especially helpful for removing tangles and mats. |
| **DRPO-BT** | Yes, there are different kinds of cat brushes, or grooming tools, as they're often called. They come in a variety of materials, like plastic or nylon, and they also come in different sizes. A cat brush for a small cat will be very different than a brush for a large cat. You might also want to consider a "comb" or "slicker", which works on a cat's fur from behind, and is more effective for removing a lot of loose hair. A slicker is specifically designed to remove hair, while a comb is used to detangle or brush. Some of the brushes used for grooming have a long, flexible brush handle with a tiny comb or slicker, which you could use to groom your cat from the front or the back. |
| **Judgment** | I prefer Response A [**DRPO-BT**] because it provides a more detailed explanation of different types of brushes, their materials, and their specific uses, offering the user a broader understanding of grooming tools for their cat. |

Table 14: Sample Responses of a HH Prompt and Judgement

| | |
|---|---|
| **Prompt** | *Human*: Is snowboarding hard to learn? |
| | *Assistant*: |
| **PPO** | Snowboarding is a challenging and demanding sport, and it takes practice and dedication to master the skills required for it. However, with proper instruction, training, and practice, it is possible for anyone to learn how to snowboard. |
| **DRPO-GPM** | It can be! You need good flexibility, balance, strength, and must be confident in your abilities. That doesn't mean you can't learn though. Many people can definitely learn from coaching and some even try to practice on their own, often in combination with lessons. Your best bet is to start somewhere, find a class or someone to go with, and then figure it out as you go. You should be able to give snowboarding a shot in a few weeks. |
| **Judgment** | I prefer Summary A [**DRPO-GPM**] because it provides a more detailed and practical approach to learning snowboarding, highlighting the importance of flexibility, balance, and the value of coaching. |

