# OpenReview forum: "Doubly Robust Alignment for Large Language Models"
_NeurIPS.cc/2025/Conference — NeurIPS 2025 poster_

### Official Review · Reviewer_YVEF · 2025-06-25

**Clarity:** 3
**Significance:** 2
**Originality:** 2
**Rating:** 3
**Confidence:** 3

**Summary:**

This paper comes up with a doubly robust estimation for RLHF. By bringing the doubly robust framework into the RLHF setting, the authors proposed the DRPO algorithm. Theoretically, they showed that the algorithm is consistent whenever the estimation of $\pi_{\mathrm{ref}}$ is consistent or the estimation of the model is consistent. Additionally, they carried experiments over IMDb dataset to show that the doubly robust property indeed appears in practice.

**Questions:**

1. When $\hat{\pi}_{\mathrm{ref}} = \pi_{\mathrm{ref}}$, does the algorithm has theoretically improvement over algorithm like PPO or DPO?
2. Can you provide any form of lower bound to demonstrate that the terms in the upper bound are tight or necessary?
3. Do you have any suggestions on improving the algorithm when the coverage assumption, i.e. $\pi/\pi_{ref}\le \epsilon^{-1}$ is not satisfied?
4. When the reward model extends beyond the Bradley-Terry framework, does your algorithm still retain a meaningful upper bound or performance guarantee? If so, under what conditions?
5. This paper explains the ‘doubly robust property’ in Theorem 2 (or Theorem 5) in terms of $\|\hat{\pi}/\pi_{ref}-1\|^2*\|g^*-\hat{g}\|^2$, as it goes to zero if any one of the above terms goes to zero. I understand that DPO/PPO are two type of algorithms which let the above two terms goes to zero respectively, but by simply from this formula it’s not clear how do we do better than any of these two algorithms. It would be better if the authors can explain an algorithm (not only a framework) which can theoretically better than PPO and DPO algorithm. I will increase my score accordingly if I am convinced that such algorithm exists.

**Ethical Concerns:**

["NO or VERY MINOR ethics concerns only"]

**Final Justification:**

I think this paper has some nontrivial results in it. But the problem formulation still lacks some novelty, and no significant technical contribution is presented in this paper. Hence I deem that this paper is slightly below the bar of acceptance.

**Limitations:**

Yes

**Paper Formatting Concerns:**

No concern in paper formatting.

**Quality:**

3

**Strengths And Weaknesses:**

Strengths:
1. This paper is well-written. The proofs are correct, and the numerical experiments seem promising.
2. The problem of double robustness in RLHF is of great interest in the community, since the number of samples required by the RLHF algorithm is usually very large, hence any kind of reduction to the number of samples is useful.
3. The algorithm DRPO also enjoys the 'doubly robust' property, which is useful in designing more general algorithms for RLHF, as it can improve the upper bounds guarantee.

Weaknesses:
1. From theoretical perspective, this paper merely join together two previously studied problem: the doubly-robust algorithms for RL, and also preference-based algorithms for RL. The semi-parametric analysis is also common in doubly robust RL literatures, e.g. [1]. Hence I question the novelty of this work. I would open to discussion and improving the score if the authors could properly justify the novelty of this paper.
2. No lower bound is provided in addition to the upper bound of the DRPO algorithm. The absence of a matching or complementary lower bound leaves open the question of optimality.
3. In the description of the results, the authors hide some factors that might be large, e.g. the dependence on the bound on the reward models. It will be better if these terms can be made explicitly in the upper bound.
4. The term 'regret' seems a little bit confusing here, as regret usually refers to the culmulative performance difference terms, and the setting in this paper is not a sequential setting. It will be better if the authors can change this term into 'regret' into something like 'performance gap'.

[1] Kallus, Nathan, and Masatoshi Uehara. "Double reinforcement learning for efficient off-policy evaluation in markov decision processes." Journal of Machine Learning Research 21.167 (2020): 1-63.

---

> ### Author Rebuttal · Authors · 2025-07-31
>
> We wholeheartedly appreciate your insightful comments, which we will carefully incorporate should the paper be accepted. During the rebuttal, we have conducted extensive theoretical and empirical studies to address the concerns raised by you and other reviewers. Below, we summarize your major comments, followed by our responses.
>
> > **Technical novelty** (Weakness 1).
>
> We understand your concern regarding novelty, given that both doubly robust (DR) methods and preference-based RLHF have been well-studied individually. Before discussing our technical novelty, we would like to highlight that the connection our work establishes between these two areas itself is novel and, we believe, highly valuable. While there is extensive literature on LLM fine-tuning, no prior work has approached it from the perspective of DR or semiparametrically efficient estimation. In that sense, our proposal not only introduces a novel methodology but also opens the door for integrating other DR and more general OPE methodologies (e.g., marginalized IS, Liu et al., 2018; conditional IS, Rowland et al., 2020; more robust DR, Farajtabar et al., 2018; to mention a few) into LLM fine-tuning.
>
> Second, regarding technical novelty, even if one aims to apply DR to RLHF, this is far from straightforward. The primary challenge stems from the difference in settings: in RLHF, we do not observe scalar reward signals but instead receive pairwise human preference. Consequently, extending standard DR -- which are designed for scalar rewards -- to reward-based RLHF (see Section 2) for learning doubly robust policy values poses significant challenges. Our approach to this challenge is to shift the focus from learning policy values to directly learning preferences. By adopting the preference-based framework, we manage to apply DR to LLM fine-tuning.
>
> Finally, we highlight a unique aspect of our DR estimator that sets it apart from existing estimators. Due to our focus on pairwise preference feedback, we are able to leverage the inherent symmetry in the data: we can treat the second response as a reference and estimate the probability that the first response is preferred using a DR estimator, or alternatively, treat the first response as the reference and estimate the probability that the second response is preferred. Our analysis shows that neither estimator alone achieves semiparametric efficiency. However, averaging the two -- taking advantage of the symmetry in pairwise preferences -- does yield a semiparametrically efficient estimator. This insight explains why our proposed estimator (see Equation (8)) is constructed as the average of two estimators. This structure arises uniquely from the nature of pairwise comparison and does not appear in prior DR or efficient RL estimators.
>
> > **Improvement over PPO/DPO** (Questions 1, 4 \& 5)
>
> Thank you for raising this important point. We appreciate the opportunity to clarify how our algorithm improves upon both DPO and PPO. We structure our response as follows, discussing our improvements in three settings: (i) when the BT model holds; (ii)  when BT does not hold; (iii) where the reference policy is correctly specified. Notice that (ii) and (iii) directly address your Questions \#4 and \#1, respectively.
>
> * **When BT model holds**. Theoretically, our proposal enjoys two key properties: (i) double robustness; (ii) (semiparametric) efficiency. The first, as you noted, ensures robustness over DPO/PPO -- specifically, the error vanishes if either of two terms converges to zero, as is required by PPO or DPO individually. To quantitatively characterize our improvement over PPO and DPO, the second property -- efficiency -- comes into play.
>
>     Specifically, semiparametric efficiency means that, in the context of preference evaluation, the proposed estimator achieves the smallest possible asymptotic MSE (see Theorem 2 and Corollary 4). As for preference optimization, this efficiency is precisely characterized in Theorem 7, where we show that the suboptimality gap of our algorithms scales as the product $\|\|\widehat{\pi}\_{\textrm{ref}}/\pi\_{\textrm{ref}} -1\|\| \|\|\widehat{r}-r^*\|\|$, whereas for PPO and DPO, their gaps scale linearly with one of these terms alone:
>
>     - PPO: $\|\| \widehat{r}-r^*\|\|$
>     - DPO: $\|\| \widehat{\pi}\_{\textrm{ref}}/\pi\_{\textrm{ref}} -1\|\|$
>
>     The product structure means that the impact of each estimation error on our algorithm is **second-order**, while in PPO and DPO it is **first-order**. Consequently, when both errors converge to zero (thanks to the approximation capabilities of transformer-based architectures), our algorithm's suboptimality converges to zero at a much faster rate than that of PPO and DPO. This provides a clear quantitative theoretical improvement over both PPO and DPO.
>
>     Additionally, our concrete algorithm is detailed on Page 12 of the Appendix. It is not merely a conceptual framework.
>
> * **Beyond BT model**. Yes, our algorithm retains meaningful performance guarantees beyond the BT framework. In fact, Theorems 2 and 5, as well as Corollaries 3, 4, and 6, do not rely on the BT model assumption. When the BT model does not hold, a scalar reward function that measures the goodness of each response may not even exist, rendering its value function undefined. In this case,  we use the total preference as the performance measure. For instance, Theorem 5 upper bounds the gap in total preference between our estimated policy and the one that maximizes the total preference. These guarantees hold under the coverage (Assumption 1) and model complexity assumptions (Assumption 4).
>
>     Next, suppose the scalar reward function does exist, but human preference does not satisfy BT. For instance, $\mathbb{P}(y_1 \succ y_2|x) = g(r(y_1,x) - r(y_2,x))$ for some monotonic link function $g$ that is not a sigmoid. In this case, the BT model is violated, and DPO/PPO no longer retain their theoretical guarantees. However, under Assumptions 2 and 3, we can extend our Theorem 7 to this setting and derive a similar suboptimality bound for our algorithm. This highlights another advantage of our approach: PPO and DPO require to correctly specify the form of the link function $g$, whereas our algorithm does not.
>
> * **When $\widehat{\pi}\_{\textrm{ref}}=\pi\_{\textrm{ref}}$**. We focus on the case where the BT model does hold, since without this assumption, DPO and PPO no longer retain their theoretical guarantees. As discussed in our earlier response, the reward estimation error does not affect the suboptimality gap of our algorithm -- this term becomes zero. In contrast, PPO’s suboptimality still depends linearly on the reward estimation error.
>
> We hope this response clarifies our theoretical improvements over DPO/PPO. Please let us know if you have any follow-up questions.
>
> > Lower bound (Weakness 2 \& Question 2)
>
> We focus on Theorem 7 which provides the suboptimality gaps for PPO, DPO and DRPO. For each algorithm, its gap contains three components:
>
> * A bias term induced by KL-regularization (proportional to $\beta$);
> * A statistical complexity term of the form $\sqrt{v/n}$, which depends on the sample size $n$ and the complexity measure $v$ of the policy class;
> * A reward/reference policy estimation error term.
>
> Since the first term can be made arbitrarily small by choosing a sufficiently small $\beta$, we discuss the tightness of the second and third terms below. For the second term, our upper bounds for PPO/DPO match the lower bounds developed in Nika et al. (2024, arXiv:2403.01857), indicating their tightness. Finally, our theoretical investigation during the rebuttal reveals that  under certain settings -- e.g., when the reward is linear and the prompt distribution is multivariate Gaussian -- the suboptimality gap of PPO depends linearly on the estimation error of the regression coefficient, which itself is proportional to the reward model estimation error. Meanwhile, for DPO, when there is a constant gap between the specified and oracle reference policy, the algorithm suffers from a constant suboptimality gap that will not converge to zero. This demonstrates the tightness of the third term.
>
> Given the tightness of the established bounds, we are more confident that our DRPO's suboptimality gap -- being second-order in the estimation error -- can be much smaller than those for PPO/DPO.
>
> > Coverage assumption (Question 3)
>
> Good point. First, we can clip the IS ratio (as implemented in our experiments) to improve the algorithm's stability. Second recent works such as Huang et al. (2025, arXiv:2407.13399) propose to replace the KL divergence with a chi-square divergence. This relaxes the uniform coverage assumption to a weaker partial coverage condition -- namely, coverage is only required to hold for the optimal policy rather than for all policies. We will discuss these approaches shall our paper be accepted.
>
> > Weaknesses 3&4
>
> We will adopt the term performance gap and revise the bound to make its dependence on the magnitude of the reward function explicit.
>
> ---
>
> We hope these address your concerns, and we would be happy to discuss further if you have any follow-up questions.

---

> > ### Comment · Reviewer_YVEF · 2025-08-05
> >
> > Thank you for your response. I don't have further questions, and I will keep my score unchanged.

---

> > > ### Author Response · Authors · 2025-08-05
> > >
> > > Dear Reviewer YVEF, thank you for your reply. We are glad to hear that our responses clarify your questions and you do not have further questions. Would you please let us know if our responses have adequately addressed all your comments, or what concerns have you remained?

---

> > > > ### Comment · Reviewer_YVEF · 2025-08-05
> > > >
> > > > Thanks the author for the follow up question. I don't have further questions for the authors. However, according to the writing, novelty and contribution of this paper, I decide to maintain my score.

---

> > > > > ### Author Response · Authors · 2025-08-06
> > > > >
> > > > > Thank you for the thoughtful feedback. We would like to clarify a few aspects and would appreciate any further guidance you can provide.
> > > > >
> > > > > - **Technical novelty**
> > > > >   As outlined in our rebuttal, (i) applying standard doubly-robust (DR) techniques to RLHF is non-trivial because reward signals are absent, and (ii) our estimator is specifically designed for pairwise-preference feedback, setting it apart from existing DR methods.
> > > > >
> > > > > - **Overall contribution**
> > > > >   Our paper presents both methodological and theoretical advances. In particular, we believe the connection we draw between DR estimation and RLHF takes a first step toward incorporating broader OPE techniques into large-language-model fine-tuning.
> > > > >
> > > > > - **Writing clarity**
> > > > >   We did not see concrete writing concerns listed under “Weaknesses” (indeed, the writing was noted as a strength), but we are, of course, happy to revise any sections that could benefit from additional explanation.
> > > > >
> > > > > We would be grateful for any specific points that remain unclear and are happy to provide additional clarification.

---

> > > > > > ### Comment · Reviewer_YVEF · 2025-08-06
> > > > > >
> > > > > > Thanks to the authors for the feedback. I think this paper is a generally good paper with solid contributions and novelty, but as you may know that NeurIPS only accepts the best papers in the field, and I think that this paper is slightly below the bar of acceptance at NeurIPS, basically due to the following:
> > > > > > 1. Even though the authors mentioned that the doubly robust structure for preference-based feedback learning is different from the doubly robust learning when the reward model is present, I view this part of the contribution to be marginal and not of great technical novelty.
> > > > > > 2. No new problem setting / proving techniques are introduced in this paper. The algorithm is merely the joint between the doubly robust structure and the traditional importance sampling algorithm.
> > > > > > 3. Even though the overall writing of this paper is good. The structure of this paper at some places is a little bit confusing, e.g.  in the theoretical results, it is not very obvious to see why the proposed algorithm is better than previous algorithms like DPO / PPO. A clearer presentation of the results will be better.

---

> > > > > > > ### Author Response · Authors · 2025-08-07
> > > > > > >
> > > > > > > Thank you for the follow-up. We believe there may have been some misunderstandings regarding our paper, which we would like to take this opportunity to clarify and address your comments.
> > > > > > >
> > > > > > > 1. **Contribution.**
> > > > > > >    We respectfully argue that our contribution is far from marginal. The key challenge lies in the fact that, although applying DR to RLHF is desirable, the majority of RLHF algorithms are not amenable to such techniques. One core contribution of our work lies in the identification that the total preference objective function used in IPO provides a structure suitable for integrating DR. This insight enables, for the first time to our knowledge, the principled application of DR estimation in the context of LLM fine-tuning. Building on this, we develop a new algorithm, rather than merely applying DR to some existing algorithms.
> > > > > > >
> > > > > > >    Second, as we responded, our proposal not only introduces a theoretically grounded algorithm, but also opens the door to broader methodological innovation providing a foundation for integrating a wide range of doubly robust and general OPE techniques into LLM fine-tuning. If one were to adopt the view that our algorithm is merely a combination of two existing fields, then by the same logic, the influential double machine learning (DML) algorithm could also be seen as a mere joint between doubly robust estimation in causal inference and machine learning. Similarly, Double RL could be viewed as a mere combination of DML and RL.
> > > > > > >
> > > > > > >    Finally, as we mentioned, due to our focus on pairwise preference feedback, we are able to leverage the inherent symmetry in the data to improve the efficiency of classical DR estimators. This sets our proposal apart from existing estimators.
> > > > > > >
> > > > > > > 2. **New proving techniques.**
> > > > > > >    A key novelty of our theoretical analysis lies in the derivation of sub-optimality bounds for DPO **without relying on linearity assumptions**.
> > > > > > >
> > > > > > >    While there is extensive literature on DPO-type algorithms, their sub-optimality gaps are relatively underexplored. Some recent works (e.g., Nika et al., 2024; arXiv:2403.01857) derive such bounds under strong parametric assumptions:  (i) log-linear policy parameterizations and (ii) linear reward functions.  These assumptions simplify the analysis by allowing the sub-optimality gap to be expressed directly in terms of parameter estimation error (see their proof of Theorem 4.2).
> > > > > > >
> > > > > > >    In contrast, our analysis proceeds without such linear assumptions, which makes the derivation much more challenging. Specifically, we introduce a softmax oracle policy $\widehat{\pi}^*$ based on the true (oracle) reward function (see Line 176 in the Appendix), and decompose the sub-optimality gap into two components:
> > > > > > >    - The sub-optimality gap of $\widehat{\pi}^*$ (bounded using the margin constant $\bar{c}$ between the best and second-best policies);
> > > > > > >    - The sub-optimality between our estimated optimal policy and $\widehat{\pi}^*$ (bounded in terms of the reward estimation error).
> > > > > > >
> > > > > > >    This decomposition enables a nonparametric analysis of DPO’s sub-optimality — without relying on linearity assumptions. Refer to Appendix A.7 for full technical details.
> > > > > > >
> > > > > > > 3. **Clear presentation of results.**
> > > > > > >    We carefully structured the presentation and repeatedly highlighted the theoretical advantages of our method over DPO and PPO:
> > > > > > >    (i) In the main paper, the second bullet point of our contributions explicitly states that our algorithm achieves a smaller sub-optimality gap compared to PPO and DPO, and refers the reader directly to Theorem 7 for details.
> > > > > > >    (ii) In the final sentence of the first paragraph of Section 5, we again clearly state that Theorem 7 demonstrates the improved sub-optimality gap relative to PPO and DPO.
> > > > > > >    (iii) Figure 2 provides a visual summary of our theoretical results and explicitly highlights that Theorem 7 shows a lower sub-optimality gap for our algorithm.
> > > > > > >    (iv) Theorem 7 itself formally presents the sub-optimality gap, and the discussion that follows reinforces the comparison, stating that our gap is smaller than those of PPO and DPO.

---

> > > > > > > > ### Comment · Reviewer_YVEF · 2025-08-07
> > > > > > > >
> > > > > > > > Thanks to the authors for the detailed response. However, my overall assessment of the paper remains unchanged. I believe the work falls slightly below the acceptance threshold, and I will be maintaining my original score.

---

### Official Review · Reviewer_b762 · 2025-06-28

**Clarity:** 3
**Significance:** 3
**Originality:** 3
**Rating:** 5
**Confidence:** 4

**Summary:**

This paper adopts doubly robust estimator for the offline policy evaluation literature and apply it in the context of large language model fine-tuning and alignment. This method is useful when the either reference model or reward model is misspecified. The double robustness ensures that unless the product of misspecification rate is converging root-n rate, the estimator is consistent. Besides theoretical justifications, this paper also demonstrates the effectiveness of the method empirically.

**Questions:**

- In Line 110, it says that Y1 and Y2 are sampled from the reference policy. But there are cases where the two responses are not generated from the reference policy like Anthropic’s HH dataset? Do your results, in particular, double robustness, rely on that assumption? How to deal with the case when the reference policy is unknown?
- Typo in line 195-196? (9) should be reduced to IS estimator when setting g_hat to zero and the DM estimator when setting the IS ratio to zero?
- Assumption 4 is unlikely to be relevant for LLM where the model capacity is really large and VC dim is thus big too. While it seems fine asymptotically, because one can have bounded and fixed VC dimension and simply increase sample size. But typically, as one increases the sample size, one would use a larger model and thus larger VC dimension as well?
- I am kind of surprised my the relative poor performance of PPO here even compared to PPO. Are you using standard hyper-parameter? Or do you do the hyperparmater search for PPO yourself?
- In line 265, when do you mean by “when the BT model assumption is violated”? In that case, how does one define preference probability?

**Ethical Concerns:**

["NO or VERY MINOR ethics concerns only"]

**Final Justification:**

I am voting for acceptance because i think this is a good paper.

**Limitations:**

Experiment setting are weak. See the weakness section.

**Quality:**

3

**Strengths And Weaknesses:**

- Strengthes
    - The paper is well-organized and clearly-written.
    - The idea is novel. I personally have thought about this before as well and it is nice to see it executed here.
- Weaknesses
    - The experiment result might be week. The base model only uses 125m parameters. That’s way below standard.

---

> ### Author Rebuttal · Authors · 2025-07-31
>
> We wholeheartedly appreciate your insightful comments, which we will carefully incorporate should the paper be accepted. Below, we first respond to your concern on our experiments. We next respond to your questions.
>
> > The experiment result might be weak. The base model only uses 125m parameters. That's way below standard.
>
> We apologize for any confusion and are grateful for the chance to clarify. First, the 125M model was used only for the **preference evaluation**  in Section 6.1, where the goal is to evaluate the MSE of our proposed doubly robust preference estimator. For the **preference optimization** experiments on the HH and TL;DR datasets, we use much larger base models -- each with billions of parameters -- as discussed in Appendix C.2.
>
> Second, in response to your concern, we have conducted an additional experiment during the rebuttal period using a **7B base model** instead of 125M for the preference evaluation. The results summarized in Table A1 below again validate the double robustness property of our estimator. We are happy to replace the original 125M results with these updated results if needed.
>
> Third, we have substantially expanded our numerical study during the rebuttal by:
>
> * adding **7** new baseline algorithms for comparison;
> * using new benchmarks for evaluation.
>
> Refer to our responses to Reviewers 8Qi6 and ZHXd for details.
>
> **Table A1**: MSE of the proposed preference estimator with a 7B base model. The preference model and reference policy can be misspecified or correctly specified.
>
> sample size | 500  | 1000 | 2000 | 3000
> -|-|-|-|-
> Both correct               | 0.002212 | 0.001160 | 0.000702 | 0.000390
> Wrong preference model  |0.024942 | 0.018757 | 0.016763 | 0.016594
> Wrong reference model   |0.066897 | 0.021389 | 0.013358 | 0.008383
> Both wrong                      | 0.265155 |0.069340 | 0.043276 | 0.045954
>
> > In Line 110, it says that Y1 and Y2 are sampled from the reference policy. But there are cases where the two responses are not generated from the reference policy like Anthropic’s HH dataset? Do your results, in particular, double robustness, rely on that assumption? How to deal with the case when the reference policy is unknown?
>
> We make some clarifications. First, $Y_1$, $Y_2$ are the compared responses in the data, rather than those simulated based on our algorithms. We define the reference policy as the distribution of these responses given the prompt.
>
> Second, our proposal accommodates unknown reference policies, as is the case with the HH dataset. It does **not** require access to the oracle reference policy. In such cases, we can estimate the reference policy via SFT and plug in this estimator for preference optimization. Indeed, the double robustness property is particularly relevant in these cases: it guarantees consistency as long as either the reference policy or the reward (or preference) model is correctly estimated—not necessarily both.
>
> When estimation errors exist, our algorithm's performance gaps in Theorems 5 \& 7 depend explicitly on these estimation errors. However, due to the double robustness property, these errors enter our suboptimality bound in a **product (second-order)** form, whereas in PPO and DPO, they appear **linearly (first-order)**. As a result, when both estimation errors converge to zero -- thanks to the strong approximation capabilities of transformer-based architectures -- our algorithm’s suboptimality gap converges to zero much faster than that of PPO and DPO.
>
> > Typo in line 195-196? (9) should be reduced to IS estimator when setting $\hat{g}$ to zero and the DM estimator when setting the IS ratio to zero?
>
> Thanks for pointing it out. You are right. We will correct the typo.
>
> > Assumption 4 is unlikely to be relevant for LLM where the model capacity is really large and VC dim is thus big too. While it seems fine asymptotically, because one can have bounded and fixed VC dimension and simply increase sample size. But typically, as one increases the sample size, one would use a larger model and thus larger VC dimension as well?
>
> Excellent comment -- thank you for raising it. Before clarifying the VC dimension, we note that Assumption 4 is only required in Theorems 5, 7 and Corollary 6. It is not needed in Theorem 2 and Corollaries 3 \& 4.
>
> Second, the VC dimension in Assumption 4 can be restricted to the complexity of the policy class $\Pi$ used during fine-tuning, rather than to the entire function class represented by the underlying LLM. While LLMs do have high capacity, fine-tuning typically restricts optimization to a much smaller and more structured subset of the parameter space -- for instance, For example, fine-tuning is often performed for only a few epochs, updates may employ low-rank adaptations such as LoRA (Hu et al., 2021), and KL regularization keeps the fine-tuned policy close to the reference policy. As a result, the effective policy class can have a much smaller VC dimension than the full model class, making this assumption more practical.
>
> > I am kind of surprised my the relative poor performance of PPO here even compared to PPO. Are you using standard hyper-parameter? Or do you do the hyperparmater search for PPO yourself?
>
> Are you referring to the relatively poor performance of PPO compared to DPO, rather than PPO itself? We employed two datasets in our main paper: TL;DR and HH. Interestingly, **PPO outperforms DPO on TL;DR**, but **performs worse than DPO on HH**.
>
> To explain this, as detailed in Appendix C.2 of our paper:
>
> * For the HH dataset, we train the reference  policy via SFT using the HH dataset itself. This ensures that the reference policy is well-aligned with the data and likely well-specified.
> * For the TL;DR dataset, we use the public model `cleanrl/EleutherAI_pythia-1b-deduped__sft__tldr` as the reference policy. However, this SFT model was trained on a filtered version of the dataset, which differs from the original dataset used in our experiments (see Huang et al., 2024). As a result, this reference policy may be misspecified for TL;DR.
>
> This explains the difference in PPO's performance compared to DPO: DPO performs better on HH (where the reference policy is correctly specified), whereas PPO performs better on TL;DR (where it is not). It also aligns with our theoretical analysis, which shows that DPO's suboptimality gap depends on the estimation error of the reference policy.
>
> **Regarding hyperparameters**, we conduct our own hyperparameter search for PPO on the HH dataset.
> Specifically, as we mentioned in Appendix C.2, training PPO on the HH dataset is highly sensitive to the reward model. We experimented with different numbers of training epochs for the reward model and found that when the reward model overfits, PPO tends to suffer from policy collapse, hacking the reward model by repeating high-reward tokens. To mitigate this, we limit the reward model training to just one epoch to avoid overfitting. We then conduct a hyperparameter search over KL coefficients $\beta \in (0.05, 0.1, 0.2)$ and learning rates $\alpha\in (1\text{e-}7, 1\text{e-}6, 3\text{e-}6)$. Finally, we set $\beta=0.05$ and $\alpha=1\text{e-}7$ as it yields the most stable training performance.
>
> > In line 265, when do you mean by “when the BT model assumption is violated”? In that case, how does one define preference probability?
>
> Violation of the BT model assumption refers to settings where human preference does not follow the BT model -- the probability of preferring one response over another cannot be expressed as a sigmoid function of the difference between their scalar rewards. Moreover, in these settings, a scalar reward function that measures the goodness of a response may not even exist.
>
> Nonetheless, the preference probability remains well-defined as a function of two responses and the prompt, even though it is no longer a function of reward differences. Additionally, these preference functions can be modeled using more general preference functions (see e.g., Zhang et al., 2024), without assuming an underlying scalar reward function.
>
> Since the reward function may not exist in these settings, the value of a policy is not well-defined. As such, we use the gap in total preference as the performance measure for our algorithm (see Theorem 5 and Corollary 6).
>
> ---
> We hope this clarifies your question, and we would be happy to elaborate further if you have any follow-up questions.

---

> > ### Comment · Reviewer_b762 · 2025-08-05
> >
> > Thanks for your rebuttal. I don't have any further questions.

---

> > > ### Author Response · Authors · 2025-08-05
> > >
> > > Thank you for your reply! We also greatly appreciate your constructive feedback and comments!

---

### Official Review · Reviewer_Q4tG · 2025-06-29

**Clarity:** 3
**Significance:** 3
**Originality:** 3
**Rating:** 5
**Confidence:** 4

**Summary:**

The authors set to improve the RLHF paradigm by recognising the fact that preferences, reward models and reference policies can be specified. To that end they introduce DRPO, which introduces doubly-robust methods to RLHF. The proposed method converges to the true preference probability when either the preference model or the reference policy is correctly specified. The method is also semi-parametric, meaning it attains the smallest MSE among all regular and asymptotically linear estimators.

The method evaluates preferences over policies and is built from a direct estimator and an importance sampling estimator. The direct estimator is the preference model using an existing reward model or DPO-type estimator (which can be generalized beyond BT models). The importance estimator which evaluates the ratios of completion pairs drawn from the trained policy and reference policy. These two estimators are combined to form the DR estimator. The main contribution is the second term in eq 8, which corrects for bias introduced by misspecification of the preference model.

The authors prove a number of theoretical results, including a regret bound (which is more robust to optimization than PPO and DPO alone). The authors also prove semi-parametric efficiency and double-robustness.

Finally, the authors perform a number of experiments. The first shows that their method works by ablating the reference policy and preference specification. The second runs a head-to-head comparison of DRPO  against PPO and DPO on standard preference optimization tasks.

**Questions:**

- Can the authors please clarify in eq 8, which part of the equation addresses preference model misspecification, reward model misspecification and reference policy misspecification?
- Can the authors expand on how this method can be generalised beyond Bradley-Terry?
- Would the authors be willing to open-source the code?

**Ethical Concerns:**

["NO or VERY MINOR ethics concerns only"]

**Final Justification:**

I found the problem, theoretical analysis and experiments quite convincing. Even if the novelty might be marginal, I still think the research is useful for the community, and the analysis is thorough enough to justify acceptance. The authors addressed my concerns during the rebuttal; I stand by my score and increase my confidence.

**Limitations:**

Yes

**Quality:**

3

**Strengths And Weaknesses:**

# Strengths
- I find the paper to be interesting and novel. Introducing the importance sampling estimator is a simple yet elegant solution towards making preference optimization more robust.
-These claims are theoretically supported by the analysis on double-robustness, semi-parametric efficiency and regret analysis.
- I find the regret analysis quite convincing since the authors show that their regret is bounded through the only the product of reference/preference policy error.
- The first experiment shows how the DRPO is robust to either type of misspecification. I appreciate the use of multiple seeds and confidence intervals, which are often missing in LM research.
- I was convinced by the second experiment, which shows a practical experiment where the reference policy is trained on a different dataset.
- I find the problem setting, theoretical results and empirical results quite convincing.

# Weaknesses
- While the paper is generally well written, I got lost in the details of section 4. I was confused because the authors mention that eq 6 and 7 require correct preference and reference policies. Then, through eq 8, the authors mention that the correction term addresses misspecification of the preference model. It was unclear to me how misspecification of the reference policy is handled. I think it would be helpful at the end of section 4 to show how each part of eq 8 supports the main claims of the paper.
- no code provided.

---

> ### Author Rebuttal · Authors · 2025-07-31
>
> We wholeheartedly appreciate your insightful comments, which we will carefully incorporate should the paper be accepted. Below, we respond to your comments and questions.
>
> > No code provided. Would the authors be willing to open-source the code?
>
> We would like to clarify that we did include our code in the Supplementary Materials.
>
> Of course, we are happy to open-source our code. Indeed, we have already published our code in an anonymous GitHub repository (Due to NeurIPS policy, we are unable to include the link).
>
> > While the paper is generally well written, I got lost in the details of section 4. I was confused because the authors mention that eq 6 and 7 require correct preference and reference policies. Then, through eq 8, the authors mention that the correction term addresses misspecification of the preference model. It was unclear to me how misspecification of the reference policy is handled. I think it would be helpful at the end of section 4 to show how each part of eq 8 supports the main claims of the paper.
>
> > Can the authors please clarify in eq 8, which part of the equation addresses preference model misspecification, reward model misspecification and reference policy misspecification?
>
> We greatly appreciate the above two thoughtful comments. Following your suggestion, we plan to include a discussion at the end of Section 4 to clarify how each type of model misspecification is handled by Eq (8).
>
> To begin with, we clarify that Eq (6) and (7) correspond to the DM and IS estimators, respectively. As you pointed out, their consistency depends on the correct specification of the preference model (for DM) and the reference policy (for IS).
>
> The DR estimator combines the virtue of DM and IS by involving both the preference model and reference policy in Eq (8). Each type of misspecification is addressed as follows:
>
> * **Reference policy misspecification**. In this case, the preference model is correct (i.e., $\widehat{g}=g^\*$). The reference policy misspecification is addressed by the inclusion of the residual $Z-g^\*$ in the second term of Eq (8). Specifically, since $\mathbb{E}[Z-g^\*(X,Y^{(1)},Y^{(2)})|Y^{(1)},Y^{(2)},X]=0$, by the law of total expectation, the second term in Equation (8) -- although it involves an incorrect IS ratio -- has zero expectation by the law of total expectation. As a result, the expectation of Eq (8) reduces to that of the first term, which corresponds to the DM estimator and remains consistent under correct specification of the preference model.
>
> * **Preference model misspecification**. In this case, we have $\widehat{\pi}\_{\textrm{ref}}=\pi\_{\textrm{ref}}$, and we can rewrite Eq (8) into
>         $$\textrm{Eq}~(7)
>         +\Big[\frac{1}{2}\sum_{a=1}^2\mathbb{E}\_{y\sim \pi(\bullet|X)} [\widehat{g}(X,y,Y^{(a)})]-\frac{\pi(Y^{(1)}|X)}{2\pi_\text{ref}(Y^{(1)}|X)}\widehat{g}(X, Y^{(1)}, Y^{(2)})- \frac{\pi(Y^{(2)}|X)}{2\pi_\text{ref}(Y^{(2)}|X)}[1-\widehat{g}(X, Y^{(1)}, Y^{(2)})]\Big]$$
>     Here, the misspecification of the preference model is addressed by the inclusion of the IS ratio in the second term within the square brackets. Specifically, with a correctly specified reference policy, the second term is of mean zero. As such, the expectation of Eq (8) is the same to that of Eq (7), which corresponds to the IS estimator and remains consistent under correct specification of the reference policy.
>
> * **Reward model misspecification**. Under the BT model assumption, one can estimate the reward and plug in this estimator to estimate the preference function. If the reward model is misspecified, the resulting preference model $\widehat{g}$ is also misspecified. Nonetheless, as in the previous case, the inclusion of the IS ratio in Eq (8) corrects this error. Thus, the same logic used to address preference model misspecification applies here as well.
>
> > Can the authors expand on how this method can be generalised beyond Bradley-Terry?
>
> Absolutely. Our proposed doubly robust algorithm relies on estimating two nuisance functions: (i) the reference policy, and (ii) the preference function -- which defines the probability that one response is preferred over another, given the prompt. When the BT model assumption is violated, this preference probability can no longer be expressed as a sigmoid function of the reward difference between two responses. However, the preference probability itself remains well-defined and can still be directly estimated from data.
>
> For example, general preference models (e.g., Zhang et al., 2024) model preferences without assuming the existence of an underlying scalar reward function. Other models such as PairRM also provide valid estimates of pairwise preference probabilities without relying on the BT assumption. These models can be used to estimate the preference probability, which can be plugged into our total preference estimator for the subsequent preference optimization.
>
> From a theoretical perspective, without the BT model assumption, a scalar reward function that measures the goodness of individual responses may not exist, rendering the policy value undefined. To address this, we use the gap in total preference as the measure to gauge the theoretical performance of our algorithm (see Theorem 5 \& Corollary 6). If a reward function does exist but the preference probabilities follow a general (non-sigmoid) link function, our Theorem 7 can be extended to derive similar suboptimality bounds in terms of policy values. We elaborate on this in our Response \#2 to Reviewer YVEF (see **Beyond BT model** paragraph).
>
> ---
> We hope this clarifies your question, and we would be happy to elaborate further if you have any follow-up questions.

---

> > ### Comment · Reviewer_Q4tG · 2025-08-03
> >
> > Thank you for these clarifications. I will stick with my current score, but increase my confidence.

---

> > > ### Author Response · Authors · 2025-08-04
> > >
> > > Thank you very much! We also greatly appreciate your constructive feedback and comments!

---

### Official Review · Reviewer_ZHXd · 2025-06-30

**Clarity:** 3
**Significance:** 2
**Originality:** 3
**Rating:** 4
**Confidence:** 3

**Summary:**

The paper proposes a doubly robust preference optimization algorithm for aligning large language models with human preferences through reinforcement learning from human feedback. The algorithm remains consistent when either the preference model or the reference policy is correctly specified, addressing model misspecification issues in existing methods. It introduces a robust and efficient preference evaluation estimator and develops a corresponding optimization algorithm for LLM fine-tuning, demonstrating superior performance in both theory and practice.

**Questions:**

- Expand Related Work: Provide a detailed comparison with recent robust DPO methods.
- Enhance Baseline Comparisons: Include more advanced and similar methods in the experimental evaluations.
- Expand Experimental Scope: Conduct experiments on additional datasets and benchmarks to validate generalizability and robustness.

**Ethical Concerns:**

["NO or VERY MINOR ethics concerns only"]

**Final Justification:**

The authors effectively addressed the core concerns through expanded related work (7 additional DPO algorithms), enhanced experimental evaluation (AlpacaEval 2.0), and clear theoretical positioning. While minor clarity issues remain regarding method scope and baseline positioning, the substantial improvements in technical rigor and comprehensive comparisons outweigh these concerns, warranting a 1-point score increase.

**Limitations:**

The authors have discussed the limitations in their manuscript.

**Quality:**

2

**Strengths And Weaknesses:**

### Strengths
1. **Innovative Algorithmic Contribution**: The paper introduces a novel doubly robust preference optimization algorithm that addresses a significant challenge in RLHF—model misspecification. This algorithm is designed to remain consistent when either the preference model or the reference policy is correctly specified, which is a substantial advancement over existing methods that often fail under such misspecifications.
2. **Theoretical Rigor and Empirical Validation**: The authors provide a thorough theoretical analysis of their proposed algorithm, establishing its double robustness and semi-parametric efficiency.

### Weaknesses
1. **Insufficient Related Work Discussion**: The paper's discussion of related work is somewhat limited, particularly in the context of robust DPO methods. Many recent advancements in robust DPO, such as "Provably Robust DPO: Aligning Language Models with Noisy Feedback" (ICML 2024), "Towards Robust Alignment of Language Models: Distributionally Robustifying Direct Preference Optimization" (ICLR 2025), and "ROPO: Robust Preference Optimization for Large Language Models" (ICLR 2025), are not adequately compared or analyzed. This omission makes it difficult to fully assess the novelty and comparative advantages of the proposed method.
2. **Weak Baseline Comparisons**: The choice of baselines in the experiments is relatively weak. The paper could benefit from comparing against more similar and advanced methods, such as the aforementioned robust DPO techniques. This would provide a more comprehensive evaluation of the proposed algorithm's performance and robustness in the context of existing state-of-the-art solutions.
3. **Limited Experimental Scope**: The experimental evaluations, while demonstrating the algorithm's effectiveness, are somewhat limited in scope. The paper could be strengthened by including more comprehensive benchmark analyses, such as using datasets like AlpacaEval2 or Arena-Hard. These additional benchmarks would provide a more thorough assessment of the algorithm's performance across a wider range of scenarios and challenges, thereby enhancing the paper's overall impact and significance.

---

> ### Author Rebuttal · Authors · 2025-07-31
>
> We wholeheartedly appreciate your insightful comments, which raise three concerns about (i) insufficient related work discussion (Weakness 1 \& Question 1); (ii) weak baseline comparisons, including the lack of robust DPO algorithms and other advanced or closely related algorithms (Weakness 2 \& Question 2) and (iii) limited experimental evaluations. In response, during the rebuttal, we have:
>
> * Provided a detailed discussion of the connections and distinctions between our doubly robust algorithm and existing robust DPO methods. We plan to incorporate this discussion into the paper shall it be accepted.
> * Added comparisons with **seven** additional DPO-style algorithms to further demonstrate the competitiveness of our proposed DRPO.
> * Employed AlpacaEval 2 to strengthen our evaluation by benchmarking across different algorithms.
>
> We next elaborate on these points below.
>
> **Insufficient related work discussion**. We appreciate you pointing out these references on robust DPO, and we fully agree that they are relevant to our work and worthwhile to cite. We have also compared against these algorithms empirically (refer to our next response). Shall our paper be accepted, we plan to use the extra page to expand the discussion section to further elaborate on the connections and differences between our work and these approaches, as follows:
>
> Recent studies have developed robust variants of DPO (Chowdhury et al., 2024; Liang et al., 2024; Wu et al., 2025), focusing on distributional robustness -- that is, robustness to discrepancies between training and test distributions. In particular, these methods studied pairwise noise, where preference labels may be flipped in the training data. In contrast, our algorithm does not address distributional shift or pairwise noise. Instead, it focuses on model misspecification and is designed to remain consistent under misspecification of the preference model or reference policy, assuming the training and test data share the same distribution.
>
> Thus, the two lines of work are robust in different senses. Our newly conducted numerical study further highlights this distinction (see our response to your next comment). Specifically, robust DPO methods tend to perform better or comparably when the level of pairwise noise or distributional shift is high, whereas our method outperforms them when such issues are mild or absent.
>
> That said, our DRPO framework is general and can be extended to handle pairwise noise: for instance, one could incorporate reward functions learned via robust DPO methods into our total preference estimator for preference optimization, or apply the weighting function $w$ from Dr. DPO to reweight each observation, in order to handle pairwise noise.
>
> If the reviewer has additional related works to suggest, we would be glad to compare and discuss them shall our paper be accepted.
>
> **Experiments**. Following your suggestion, we compared against **seven** baseline algorithms during the rebuttal. These include the robust DPO methods **rDPO** and **Dr. DPO** that you mentioned, as well as **cDPO** (Mitchell, 2023). We also included more advanced DPO-type algorithms such as **RSO** (Zhao et al., 2023) and **CPO** (Xu et al., 2024). Finally, we included **IPO** (Azar et al., 2024)   and **ORPO** (Hong et al., 2024), which are two closely related algorithms to ours. Most algorithms were implemented using Hugging Face’s `trl` library. An exception is DR-DPO, which we trained using a modified version of the DPO trainer based on the authors’ released code, as its official `trl` implementation is not available. Unfortunately, due to the lack of publicly available code for **ROPO**, we were unable to include it in our experimental comparison at this time.
>
> Table B1 below reports the win rate and length-controlled win rate of different algorithms relative to the SFT policy on the HH dataset. Both win rates are evaluated using **AlpacaEval 2.0** you suggested. Table B2 reports the win rate of the proposed DRPO method compared to the seven baseline algorithms on the TL;DR dataset.
>
> On the HH dataset, DRPO achieves higher win rates than DPO-style algorithms (**RSO**, **CPO**) as well as **IPO** and **ORPO**, performs comparably to robust DPO methods **rDPO** and **cDPO**, and underperforms **Dr. DPO**. On the TL;DR dataset, however, DRPO consistently achieves win rates above 50% across all comparisons—often exceeding **60\%** and in some cases **90\%**.
>
> This performance difference likely arises from the presence (or absence) of pairwise noise in the datasets. DRPO is specifically designed to handle model misspecification rather than noisy labels. Its strong performance on TL;DR suggests that this dataset contains minimal pairwise noise—making DRPO more effective than methods tailored to robustness under label noise. In both the rDPO and Dr. DPO papers, the authors found that setting the flip probability to $\epsilon = 0.1$ yields the best performance on the HH dataset, suggesting that HH contains certain levels of pairwise noise. We adopt the same $\epsilon$ in our experiments to ensure these algorithms attain their best performance. These pairwise-noise-robust methods thus outperform or match DRPO on HH, as expected.
>
> Finally, we note that TL;DR is a domain-specific summarization task. General-purpose benchmarks such as AlpacaEval 2.0 may not be appropriate for the evaluation on this dataset. Therefore, we follow the evaluation protocol from DPO and P3O to assess the quality of the generated responses from each algorithm (Note that rDPO and Dr. DPO did not report results on this dataset).
>
> **Table B1**. Win rate and length-controlled win rate of different algorithms relative to the SFT policy on the HH dataset. Higher win rates indicate better performance.
>
> | Model    | LC Win Rate (%) | Win Rate (%) |
> |----------|------------------|---------------|
> | DPO      | 83.90            | 84.09         |
> | Dr. DPO  | 92.16            | 90.93         |
> | rDPO     | 86.89            | 85.71         |
> | cDPO     | 85.05            | 84.28         |
> | CPO      | 73.59            | 71.28         |
> | ORPO     | 75.92            | 53.91         |
> | IPO      | 78.29            | 78.88         |
> | RSO      | 80.62            | 79.50         |
> | DRPO     | 86.38            | 84.84         |
>
> **Table B2**. Win rate of the proposed DRPO method compared to various baseline algorithms on the TL;DR dataset. Higher win rates indicate better performance of DRPO over the baseline algorithm.
>
> | Additional Baseline Model | Win Rate (%) |
> |---------------------------|--------------|
> | DRPO vs Dr. DPO           | 72.5         |
> | DRPO vs rDPO              | 65.0         |
> | DRPO vs cDPO              | 63.5         |
> | DRPO vs CPO               | 90.0         |
> | DRPO vs ORPO              | 57.5         |
> | DRPO vs IPO               | 98.5         |
> | DRPO vs RSO               | 69.5         |
>
> ---
>
> We hope these address your concerns, and we would be happy to discuss further if you have any follow-up questions.

---

> > ### Author Response · Authors · 2025-08-05
> >
> > Dear Reviewer ZHXd, once again, we greatly appreciate your constructive comments and feedback. Would you please let us know if our responses have adequately addressed your comments, or if you have any follow-up questions?

---

> > ### Comment · Reviewer_ZHXd · 2025-08-06
> >
> > The authors have addressed the major concerns through expanded related work discussion, additional baseline comparisons with 7 DPO algorithms, and enhanced experimental evaluation using AlpacaEval 2.0. However, the paper would benefit from clearer articulation of DRPO's specific applicability scope and more explicit positioning relative to other robust DPO baselines to enhance clarity for readers.
> >
> > I will increase score by 1 point based on these substantial improvements, with the recommendation to further clarify method scope and baseline positioning.

---

> > > ### Author Response · Authors · 2025-08-06
> > >
> > > Dear Reviewer ZHXd, we are glad that our rebuttal has addressed your major concerns, and we sincerely appreciate your thoughtful reconsideration and the improved score. Shall our paper be accepted, we plan to clarify in the related work and discussion sections our scope and positioning relative to other robust DPO baselines, as you recommended. Once again, we are extremely grateful for your insightful comments and constructive suggestions.

---

### Official Review · Reviewer_8Qi6 · 2025-07-03

**Clarity:** 2
**Significance:** 3
**Originality:** 3
**Rating:** 4
**Confidence:** 2

**Summary:**

The paper leverages the characteristic that preference learning consistency is maintained when either preference modeling or reference policy is correctly specified, proposing a novel preference alignment algorithm, DRPO, which uses a doubly robust learning approach that considers both simultaneously. Experiments were conducted on IMDb, TL;DR, and Anthropic Helpful and Harmless datasets, achieving superior performance compared to DPO.

**Questions:**

1. question whether this approach scales effectively to larger models and commonly used preference learning datasets.

**Ethical Concerns:**

["NO or VERY MINOR ethics concerns only"]

**Final Justification:**

My primary concern was the absence of a baseline, which the reviewer addressed by conducting seven additional baseline experiments.
Additionally, experiments on the robustness of model size resolved my related doubts.
Regarding concerns about the experimental data, I determined it was not a significant issue, as the data is widely used in alignment research.

As a result, I have increased the score by 1 point.

**Limitations:**

yes

**Quality:**

3

**Strengths And Weaknesses:**

Strengths
1. Decomposing the misspecification issue into preference modeling and reference policy problems and tackling them jointly is a novel approach.
2. The core idea underlying this research's methodology, which consistency is maintained when either preference modeling or reference policy is correctly specified, has been verified both mathematically and through toy-scale experiments, providing evidence for the validity of this approach.
3. The ablation studies on the main objective functions used in the experiments are well-conducted.

Weaknesses
1. I question whether the experimental setup is appropriate for demonstrating preference model misspecification and reference model misspecification. The IMDb sentiment, TLDR Reddit, and HH datasets have very low preference complexity, with pos-neg pair compositions that are easily distinguishable. The results obtained from such setups are inadequate for demonstrating the resolution of preference misspecifications, which is the main motivation of this research. This paper should be experience on datasets containing diverse human preferences like UltraFeedback, or in setups with mixed preferences within datasets as in the Dr. DPO paper (Which research target to robust preference alignement).
2. The experiments were conducted on models under 1B parameters with relatively low-difficulty datasets. It is questionable whether the method will scale well to commonly used models around 7B parameters and widely adopted preference learning datasets like UltraFeedback.
3. Given that this work targets robust alignment, there should be relevant baselines for comparison, but they are absent. Comparisons with methods like IPO or Dr. DPO should be included.

---

> ### Author Rebuttal · Authors · 2025-07-31
>
> We wholeheartedly appreciate your insightful comments. During the rebuttal, we have conducted extensive experiments to strengthen our empirical evaluation and address your concerns. In summary, we have: (i) scaled up the model size; (ii) included **seven** more recent baseline algorithms for comparison; and (iii) adopted more comprehensive benchmark protocols for evaluation. We next elaborate on these points. Finally, we discuss the appropriateness of our experimental setup for demonstrating preference model misspecification and reference model misspecification.
>
> > Weakness 3: Given that this work targets robust alignment, there should be relevant baselines for comparison, but they are absent. Comparisons with methods like IPO or Dr. DPO should be included.
>
> We appreciate you pointing out these robust alignment algorithms such as Dr. DPO, and we fully agree that they are relevant to our work and have compared them empirically during the rebuttal. Before elaborating on our experiments, we first clarify our proposed DRPO and robust alignment methods like Dr. DPO address different forms of robustness. Specifically, algorithms such as Dr. DPO are **robust to the discrepancies between training and test distributions**, such as pairwise noise in the training data. In contrast, our DRPO is **robust to the model misspecification** (misspecification of the preference model or reference policy), **assuming the training and test data share the same distribution**. Our experiments, detailed below, also highlights this difference.
>
> To address your comment, we compared against **seven** baseline algorithms during the rebuttal. These include the robust DPO methods such as **Dr.DPO** that you mentioned, as well as **cDPO** (Mitchell, 2023) and **rDPO** (Chowdhury et al., 2024). We also included more advanced DPO-type algorithms such as **RSO** (Zhao et al., 2023) and **CPO** (Xu et al., 2024). Finally, we included **IPO** that you mentioned, and **ORPO** (Hong et al., 2024), another recent preference optimization algorithm. Most algorithms were implemented using Hugging Face’s `trl` library. An exception is Dr. DPO, which we trained using a modified version of the DPO trainer, as its official `trl` implementation is not available.
>
> Table A1 below reports the win rate and length-controlled win rate of different algorithms relative to the SFT policy on the HH dataset. Both win rates are evaluated using AlpacaEval 2.0 (following the suggestion from Reviewer ZHXd). Table A2 reports the win rate of the proposed DRPO method compared to the seven baseline algorithms on the TL;DR dataset.
>
> On the HH dataset, DRPO achieves higher win rates than DPO-style algorithms (**RSO**, **CPO**) as well as **IPO** and **ORPO**, performs comparably to robust DPO methods **rDPO** and **cDPO**, and underperforms **Dr. DPO**. On the TL;DR dataset, however, DRPO consistently achieves win rates above 50% across all comparisons—often exceeding **60\%** and in some cases **90\%**.
>
> This performance difference likely arises from the presence (or absence) of pairwise noise in the datasets. DRPO is specifically designed to handle model misspecification rather than noisy labels. Its strong performance on TL;DR suggests that this dataset contains minimal pairwise noise—making DRPO more effective than methods tailored to robustness under label noise. In both the rDPO and Dr. DPO papers, the authors found that setting the flip probability to $\epsilon = 0.1$ yields the best performance on the HH dataset, suggesting that HH contains certain levels of pairwise noise. We adopt the same $\epsilon$ in our experiments to ensure these algorithms attain their best performance. These pairwise-noise-robust methods thus outperform or match DRPO on HH, as expected. Meanwhile, it is also possible to integrate these robust alignment algorithms with our DRPO to enhance its performance in handling pairwise noise: for instance, one could incorporate reward functions learned via robust DPO methods into our total preference estimator for preference optimization, or apply the weighting function $w$ from DR. DPO to reweight each observation, in order to handle pairwise noise.
>
> **Table A1**. Win rate and length-controlled win rate of different algorithms relative to the SFT policy on the HH dataset. Higher win rates indicate better performance.
>
> | Model    | LC Win Rate (%) | Win Rate (%) |
> |----------|------------------|---------------|
> | DPO      | 83.90            | 84.09         |
> | Dr. DPO  | 92.16            | 90.93         |
> | rDPO     | 86.89            | 85.71         |
> | cDPO     | 85.05            | 84.28         |
> | CPO      | 73.59            | 71.28         |
> | ORPO     | 75.92            | 53.91         |
> | IPO      | 78.29            | 78.88         |
> | RSO      | 80.62            | 79.50         |
> | DRPO     | 86.38            | 84.84         |
>
> **Table A2**. Win rate of the proposed DRPO method compared to various baseline algorithms on the TL;DR dataset. Higher win rates indicate better performance of DRPO over the baseline algorithm.
>
> | Additional Baseline Model | Win Rate (%) |
> |---------------------------|--------------|
> | DRPO vs Dr. DPO           | 72.5         |
> | DRPO vs rDPO              | 65.0         |
> | DRPO vs cDPO              | 63.5         |
> | DRPO vs CPO               | 90.0         |
> | DRPO vs ORPO              | 57.5         |
> | DRPO vs IPO               | 98.5         |
> | DRPO vs RSO               | 69.5         |
>
> > Weakness 2 and Question 1: The experiments were conducted on models under 1B parameters. I question whether this approach scales effectively to larger models.
>
> We apologize for any confusion. First, in the main paper, we clarify that only the **preference evaluation** experiments in Section 6.1 were conducted with models with less than 1B parameters. For the **preference optimization** experiments on the HH and TL;DR datasets, we use larger base models -- each with billions of parameters -- as discussed in Appendix C.2.
>
> Second, to directly address your concern, we have conducted an additional experiment during the rebuttal period using a **7B base model** for the preference evaluation. The results summarized in Table A3 below again validate the double robustness property of our estimator. We are happy to replace the original results with these updated results shall our paper be accepted.
>
> **Table A3**: MSE of the proposed preference estimator with a 7B base model. The preference model and reference policy can be misspecified or correctly specified.
>
> Sample size | 500  | 1000 | 2000 | 3000
> -|-|-|-|-
> Both correct               | 0.002212 | 0.001160 | 0.000702 | 0.000390
> Wrong preference model  |0.024942 | 0.018757 | 0.016763 | 0.016594
> Wrong reference model   |0.066897 | 0.021389 | 0.013358 | 0.008383
> Both wrong                      | 0.265155 |0.069340 | 0.043276 | 0.045954
>
>
> > Weakness 1: Whether the experimental setup is appropriate for demonstrating preference model misspecification and reference model misspecification.
>
> We demonstrate the appropriateness of our employed experimental datasets for investigating preference model misspecification and reference model misspecification below.
>
> First, **the HH dataset is well-suited for illustrating preference model misspecification**. As mentioned earlier, both the rDPO and Dr. DPO papers noted that this dataset likely contains unknown pairwise noise that is not captured by standard preference
> (e.g., BT) -- thus violating the assumed preference model. Indeed, the rDPO paper deliberately avoids flipping preferences in this dataset due to the suspected presence of inherent noise, and finds that setting the flip probability to $\epsilon = 0.1$ yields optimal performance. The Dr. DPO paper similarly reports improved results when applying a flip rate of 0.1, even when the dataset itself is not explicitly flipped. These observations indicate that the HH dataset contains mixed preferences -- a setting you recommended to conducted experiments -- making it appropriate for testing the impact of preference model misspecification.
>
> Second, **the TL;DR dataset is well-suited for illustrating reference model misspecification**. As detailed in Appendix C.2, we use the public model `cleanrl/EleutherAI_pythia-1b-deduped__sft__tldr` as the reference policy. However, this SFT model was trained on a filtered version of the dataset, which differs from the full TL;DR dataset used in our experiments (see Huang et al., 2024). As a result, the reference policy may be misspecified for our setting. Our experimental findings support this: we observe that DPO, which requires a correctly specified reference policy for consistency, underperforms PPO, suggesting potential reference model misspecification.
>
> Finally, **the IMDb dataset serves as a useful illustrative example for investigating both types of misspecification**. As it is a synthetic dataset, we have access to the ground-truth preference and reference models, allowing us to precisely control model misspecification. Notably, both the rDPO and Dr. DPO papers also use this dataset. For instance, rDPO introduces pairwise noise to simulate preference model misspecification, while in our setup, we intensify this by randomly assigning preferences (each response being preferred with 50% probability). Consequently, our simulated data has even larger preference complexity than in rDPO.
>
> ---
>
> We hope these address your concerns, and we would be happy to discuss further if you have any follow-up questions.

---

> > ### Comment · Reviewer_8Qi6 · 2025-08-05
> >
> > Thank you for your response. Many of my questions have been resolved, and I will increase the score.

---

> > > ### Author Response · Authors · 2025-08-05
> > >
> > > Dear Reviewer 8Qi6, we're glad to hear that our responses have addressed your questions, and we greatly appreciate your raising the score. Once again, we sincerely thank you for your constructive comments and feedback.

---

### Decision · Program_Chairs · 2025-09-17

**Decision:**

Accept (poster)

**Comment:**

This paper adapts the classical doubly robust estimator technique to the problem of offline alignment in RLHF. The main result is a new RLHF algorithm that achieves double robustness with respect to the preference model and the reference/data collection policy. The authors provide a sample complexity analysis for the algorithm, as well as an empirical validation.

Reviewers initially had some reservations, but were positive after the discussion period, particularly because the authors were able to include more extensive experimental results. Overall, the paper is competently executed and well written, and is likely to be of interest to the RLHF community